# SWE-EVAL: TRAJECTORY-ENHANCED EVALUATION FOR AGENTIC ISSUE RESOLUTION

## ABSTRACT

Agents and Language Models (LMs) demonstrate significant advancements in software engineering, particularly in issue resolution. Current benchmarks can qualitatively assess the correctness of generated patches. However, they lack mechanisms for quantitatively evaluating the trajectory, which is important to reveal the point of improvement. To obtain understanding of issue-resolving agents' working processes, we propose SWE-eval, a trajectory-augmented evaluation framework. SWE-eval additionally assesses a coding agent's reasoning trajectory across three dimensions: **(1) Efficiency**, measured by resource consumption; **(2) Logical Consistency**, where *Intra-turns* measures the logical consistency within a single turn and *Inter-turns* measures logical consistency across multiple conversation turns; **(3) Tool Utilization**, for which we design a metric *Info-gain* to assess how much new information the tool provides for solving problems. Our experiments on three agents and nine LMs demonstrate that SWE-eval effectively reveals underlying interpretations of agent performance and can guide development of more effective agents. First, our evaluations show that elevating trajectory-aware metrics is crucial for enhancing the *% Resolved*. Second, we trace divergent agent behaviors to shallow exploration, missing backtracking, and loop entrapment. We also show that fine-tuning on agents risks overfitting and scaling LMs improves trajectories. Third, LLM-based evaluations align closely with expert judgments and exhibit consistent stability, serving as reliable proxies.

## 1 INTRODUCTION

Language Models (LMs) are increasingly employed in building software engineering agents. The effectiveness of agents (Wang et al., 2025; Yang et al., 2024b) is rigorously evaluated on standard benchmarks such as HumanEval (Yadav & Mondal, 2025), MBPP (Austin et al., 2021), and SWE-bench (Jimenez et al., 2023). Among various benchmarks (Li et al., 2024a;b) that assess coding agents, SWE-bench and its derivatives (Pan et al., 2025; Kio, 2024; Zhang et al., 2025a; OpenAI, 2024) focus on issue resolution task, which best mirrors complex real-world development. This task involves taking a issue description and a codebase as input and generating a corresponding patch.

Despite recent progress, most benchmarks measure only patch correctness, overlooking critical trajectory-aware analysis (i.e., analysis of multi-turn conversation). This gap obscures how solutions are derived and why they fail, hindering rigorous diagnosis and interpretability. Trajectory evaluation faces three key challenges: (1) trajectories have complex structures, combining long code and natural language. (2) they involve intricate logic, requiring advanced semantic analysis. (3) agents use diverse tools and strategies. As a result, simple methods are inadequate for evaluating long, semantically complex, and varied trajectories.

We present two examples in Figure 1 to illustrate how trajectory analysis reveals specific shortcomings of coding agents. As shown in Figure 1b, SWE-agent (Yang et al., 2024a) reproduces issue first . But it exhausts turns limitation due to poorly tools utilization, which prevents generation of a final patch. As for the trajectory shown in Figure 1a, Moatless (aorwall, 2025) repeats the same error for 12 turns. Despite the Large Language Models (LLMs) explicitly acknowledging, "I realize that I'm repeatedly making the same mistake," it fails to self-correct. This persistent, self-acknowledged erroneous behavior necessitates robust *Stuck-in-Loop* recovery mechanisms.

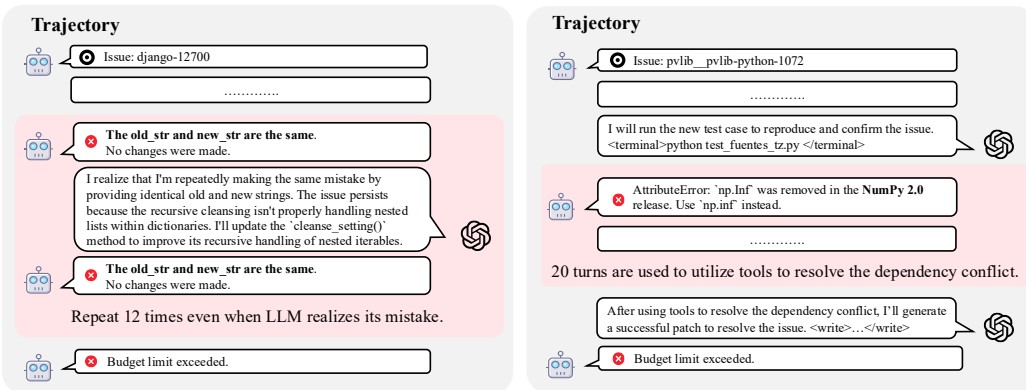

(a) Case of poor logical consistency.      (b) Case of poor tool utilization.

Figure 1: Look into the trajectory of unresolved issues. (a) Dependency conflict are identified while testing newly generated test cases. (b) Get stuck in a loop despite recognizing mistakes.

To address these limitations, we introduce SWE-eval, a trajectory-augmented evaluation framework, which goes beyond **Patch Correctness**. As illustrated in Figure 2, we evaluate three trajectory-aware dimensions: (1) **Efficiency**: we track *# Tokens* and *# Turns* to quantify resource usage and interaction cost; (2) **Logical Consistency**: we introduce a rule-based *% Stuck-in-Loop* detector with two LLM-based checks. *Inter-turn* tests each response against the task specification. *Intra-turn* detects contradictions within a single response; (3) **Tool Utilization**: we use rule-based *% Tool Suc.* to measure call reliability, and LLM-based *Info-gain* to estimate the marginal information added by each turn, isolating the contribution of tool interactions. Together, these metrics provide a multi-granular diagnosis of causal factors, explaining why reasoning succeeds or fails.

Experiments with three agents and nine LMs show that SWE-eval effectively reveals the mechanisms underlying success and failure in agentic issue resolution. By comparing resolved and unresolved trajectories, we find that success is associated with fewer turns, reduced *Stuck-in-Loop*, and improved *% Tool Suc.*, *Info-gain*, *Inter-turns*, and *Intra-turns*. Comparative analysis delineates distinct failure modes: SWE-agent exhibits shallow exploration; OpenHands lacks backtracking, limiting error recovery; Moatless is prone to loop entrapment. LMs evaluation shows systematic effects: OpenHands-specific fine-tuning overfits tool schemas and reduces SWE-agent tool use (from 31.84 to 22.20); scaling increases resolution rates (from 2.68% to 31.7%) and lowers *Stuck-in-Loop* incidence (from 53.85% to 8.79%). Furthermore, we validate the reliability (ICC up to 0.81 against experts) and and consistency (Mean diff up to 0.07) of LLM-based evaluations. Finally, we present a Django-12700 case study exposing repetitive error loops and oversized patches, underscoring the need for trajectory-aware evaluation to advance robust agent design.

Our main contributions are summarized as follows: (1) We introduce SWE-eval, a trajectory-augmented evaluation framework that moves beyond patch correctness. SWE-eval extends evaluation along three trajectory-aware dimensions: Efficiency, Logical Consistency, and Tool Utilization. (2) We perform a quantitative evaluation of three agents and nine LMs, providing both performance scores and the corresponding rationale. Our analysis uncovers the mechanisms driving performance differences, and highlights unresolved failure modes that indicate potential directions for future improvement. (3) We show that SWE-eval show strong alignment with human ratings, displaying a reasonable distribution. The scores are stable and consistent through repeated evaluations.

## 2 SWE-eval

### 2.1 Task Definition

**Evaluating Trajectory and Patch**    We evaluate trajectory from three critical aspects: Efficiency, Tool Utilization, and Logical Consistency. Efficiency involves analysis of resource consumption throughout the agent's operation. Tool utilization quantifies how effectively each tool invocation

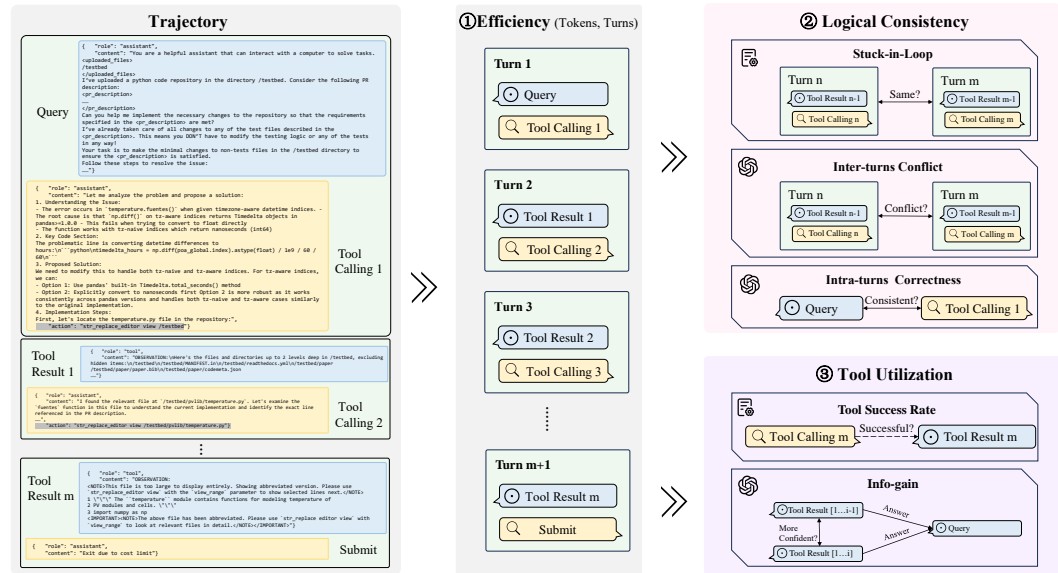

Figure 2: SWE-eval, a trajectory-augmented evaluation framework for issue resolution. Beyond accessing patch correctness, SWE-eval conducts trajectory-augmented evaluation along three additional dimensions: Efficiency, Logical Consistency, and Tool Utilization.

advances the agent towards the final solution. Logical Consistency involves evaluating the validity of individual reasoning steps and the coherence of the overall sequence.

## 2.2 TRAJECTORY EVALUATION

Trajectory-aware metrics are designed to find the reason of unresolved issue. As shown in Figure 2, SWE-eval is a trajectory-augmented multidimensional evaluation framework for agent-driven GitHub issue resolution. While prior work has largely focused on assessing patch correctness, SWE-eval advances the evaluation frontier by conducting trajectory-level assessment along three additional dimensions: Efficiency, Logical Consistency and Tool Utilization. The detailed description of each dimension is provided in the following subsections.

**Efficiency** The evaluation of efficiency aims to quantify the resources consumed by the agent while solving a task. In software engineering contexts, an ideal agent should not only successfully complete the task but also do so in the most economical way. This directly relates to its deployment cost and practical viability in real-world scenarios.

(1) The number of tokens (**# Tokens**) is directly counted from the official interface. This method ensures accuracy and reproducibility due to the interface's authoritative nature. Different Large Language Models (LLMs) tokenize text in different ways.

(2) The number of tokens (**# Turns**) is defined as the numbers of interaction cycle. While the initial input query originates from a user, all subsequent input messages are autonomously generated by the agent itself. Each turn $i$ consists of an input query $q_i$ to the LLM and its response $r_i$. The initial query $q_1$ packages the system prompt and task description. For $i > 1$, $q_i$ provides the result of the preceding tool invocation. Each response $r_i$ issues exactly one tool call to enable deterministic execution. To formalize sequential interactions, we define turn context $t_i$ as follows:

$$t_i = q_i + r_i \tag{1}$$

**Logical Consistency** Evaluating the logical consistency of a trajectory is intended to analyze whether the agent's "chain of thought" is coherent and rational. An agent that ultimately solves a problem through a process fraught with contradictions or inefficient loops demonstrates an unstable and less interpretable solution strategy. Such an agent may exhibit unpredictable behavior when

faced with novel problems. This dimension, therefore, reflects the robustness and reliability of the agent's reasoning capabilities.

(1) The percentage of Stuck-in-Loop (**% Stuck-in-Loop**) is percentage of continuous repetitions of identical actions, which is frequently observed in open-source LMs (Pan et al., 2025). Quantifying this phenomenon therefore helps assess a model's interaction stability and task execution capabilities. We define a *Stuck-in-Loop* state when identical response text appears at least three times across turns. We detect it with a **hash-based counter**: the key is the exact LLM response string $r_i$ and the value is its occurrence count. This captures cross-turn repetition, including non-consecutive repeats.

(2) Inter-turns Conflict (**Inter-turns**) measure whether a turn maintains consistency with all prior information, ensuring no contradictions. We define $S(t_n; t_m)$ as the LLM's assessed consistency between $t_n$ and $t_m$:

$$\text{Inter-turns}_i = S(t_i, \sum_{m=1}^{i-1} t_m) \tag{2}$$

(3) Intra-turns Correctness (**Intra-turns**) evaluates response rationality to follow user instructions within a turn. At turn $i$, the LLM outputs response $r_i$ to query $q_i$. We use the subsequent user feedback $q_{i+1}$ as implicit supervision to assess the correctness of $r_i$. $C(q_i, r_i|q_{i+1})$ denotes the Judge LLM's confidence score for response $r_i$ effectively satisfying query $q_i$, given feedback $q_{i+1}$. Formally, this *Intra-turn* evaluation is expressed as:

$$\text{Intra-turns}_i = C(q_i, r_i \mid q_{i+1}) \tag{3}$$

**Tool Utilization** In software engineering tasks, coding agents heavily rely on interactions with external tools (e.g., compilers, file systems, debuggers) to perceive the state of the environment and execute modifications. Therefore, tool utilization is not only a key measure of their behavioral effectiveness but also a direct reflection of their core ability to understand tool semantics, formulate effective plans, and engage in meaningful interactions with the environment.

(1) Tool Call Success Rate (**% Tool Suc.**) reflects the agent's ability to interact with environment, understand tool semantics. We let success($q_i$) return 1 if the tool call in turn $i$ is executed without runtime errors and produces a valid result. Let $Q = \{q_1, ..., q_k\}$ be the set of all tool invocations result in trajectory. *% Tool Suc.* is formally defined as:

$$R_{\text{success}} = \frac{\sum_{i=1}^{k} \text{success}(q_i), q_i \in Q}{|Q|} \tag{4}$$

(2) Information Gain (**Info-gain**) evaluates the utility of each tool invocation within a query resolution process. To quantify its individual contribution, *Info-gain* directly assesses the incremental knowledge a tool call contributes. Minimal *Info-gain* suggests redundancy or limited utility, highlighting areas for optimization. Formally, the initial user query is represented by $\hat{q}$. $G(t_m, \hat{q}|t_n)$ denotes the LLM's assessed information gain in accurately answering question $\hat{q}$ by incorporating the new context $t_m$, relative to relying solely on context $t_n$. We define $\sum_{m=1}^{i} t_m$ as all content from turn 1 to turn i. Specifically, *Info-gain$_i$* is formally expressed as:

$$\text{Info-gain}_i = G(t_i, \hat{q} \mid \sum_{m=1}^{i-1} t_m) \tag{5}$$

**LLM-as-a-Judge** We use LLMs in *Inter-turns*, *Intra-turns*, and *Info-gain* for two key reasons: (1) LLMs provide both quantitative scores and qualitative explanations, which are crucial for identifying issues in the SWE-eval trajectory analysis. (2) LLMs are well-suited for addressing challenges in trajectory evaluation, such as complex structures, intricate logic, and diverse agent strategies.

To ensure a robust and reliable evaluation, we employed three distinct LLMs to independently generate both the primary **score**, G/C/S, and its corresponding **rationale**. This multi-LLM approach was specifically chosen to mitigate inherent biases and idiosyncratic preferences often present in individual models. Subsequently, we aggregated these outputs by averaging the generated score and synthesizing the rationales, thereby enhancing objectivity. See Appendix C.2 for a full prompt.

### 2.3 PATCH LAYER EVALUATION

**Patch Correctness** (1) Patch Apply Rate (**% Apply**) indicates the percentage of generated patches that are syntactically correct and can be successfully applied to the codebase without errors (Jimenez et al., 2023). A low apply rate reveals problems in code syntax generation and structural correctness.

(2) Patch Resolved Rate (**% Resolved**) measures the proportion of patches successfully passes all tests (Jimenez et al., 2023), which reflects the ability of producing functionally correct patches.

## 3 EXPERIMENT SETUP

**Agents** We conduct experiments using three representative agents. These include the general-purpose coding agent **OpenHands** (Wang et al., 2025) (paired with the CodeAct agent (Wang et al., 2024)), **Moatless** which integrates Monte Carlo Tree Search with a self-improvement mechanism (aorwall, 2025), and agent specifically designed for issue-resolution tasks: **SWE-agent** (Yang et al., 2024a). We limit the number of LM calls to 30 per instance to balance computational efficiency and correctness.

**LMs** We present the performance of 2 LLMs and 7 Small Language Models (SLMs) that span different size and functional categories, as shown in Appendix C.1. By including general, reason, code, and fine-tuned models, we aim to compare different kinds LMs.

**Benchmark** Experimental results for SWE-bench-Lite and SWE-bench-Verified are presented in the main text to prioritize our core findings. SWE-bench-Lite comprises 300 instances, specifically chosen to enable rapid evaluation cycles. SWE-bench-Verified (OpenAI, 2024) was introduced by OpenAI to improve the robustness and reliability of the SWE-bench evaluation, containing a subset of 500 samples that 93 professional human annotators verified to be non-problematic (Yu et al., 2025). The distribution of different type of issues in SWE-bench-Lite are measured in the Appendix G.1. Evaluations on other benchmarks are provided in the Appendix H.

## 4 EXPERIMENT RESULTS

In this section, we report and analyze the experimental results to answer the following research questions (RQs):

- **RQ1: Does SWE-eval correlate with problem-solving success?** This research question accesses whether SWE-eval reliably separates resolved from unresolved instances. Clear separation would provide diagnostic report and actionable guidance.

- **RQ2: How effective does SWE-eval expose the limitations of agents?** We test whether SWE-eval consistently surfaces failure modes and distinguishes strengths and weaknesses across systems.

- **RQ3: How reliable and consistent is SWE-eval?** This research question investigates whether LLM-based metrics align with human judgment. It also examines the consistency of results across multiple measurements, as well as the reasonableness of the distribution.

### 4.1 RQ1: DOES SWE-EVAL CORRELATE WITH PROBLEM-SOLVING SUCCESS?

We test whether trajectory-aware metrics predict *% Resolved*. To avoid model assumptions, we use a model-agnostic comparison. We split instances into resolved and unresolved and compare their statistics. In Table 1, green denotes metrics with better values for resolved than for unresolved instances. This assesses separability without training a classifier. (1) Efficiency. Resolved instances require fewer *# Turns* and less compute, suggesting success aligns with efficient resource use. (2) Logical consistency. Resolved trajectories show lower *% Stuck-in-Loop* and higher *Inter-turns* and *Intra-turns*. The largest effects are the drop in *% Stuck-in-Loop* and the rise in *Intra-turns*, consistent with more coherent, efficient progress. (3) Tool utilization. Resolved instances use tools more effectively: higher *% Tool Suc.* indicates more successful invocations, and larger *Info-gain* shows

Table 1: Comparison between resolved and unresolved instances

| Benchmark | Agent | Resolved | Efficiency | | Consistency | | | Tool Utilization | |
|---|---|---|---|---|---|---|---|---|---|
| | | | # Tokens | # Turns | % Stuck-in-Loop | Inter-turns | Intra-turns | % Tool Suc. | Info-gain |
| SWE-bench_Lite | SWE-agent | ✓ | 44950.00 | 6.12 | 0.00 | 72.39 | 62.47 | 53.06 | 62.47 |
| | | ✗ | 31361.78 | 4.97 | 0.00 | 69.30 | 59.77 | 52.40 | 59.77 |
| | OpenHands | ✓ | 273969.07 | 15.28 | 2.47 | 74.81 | 70.77 | 59.03 | 70.77 |
| | | ✗ | 543793.22 | 22.82 | 7.34 | 74.90 | 69.56 | 53.27 | 69.56 |
| | Moatless | ✓ | 227417.85 | 20.00 | 12.20 | 76.63 | 75.46 | 71.49 | 75.46 |
| | | ✗ | 372810.27 | 26.91 | 37.31 | 75.67 | 64.82 | 68.32 | 64.82 |
| SWE-bench_Verified | SWE-agent | ✓ | 108967.81 | 10.81 | 0.00 | 74.63 | 68.70 | 67.47 | 68.70 |
| | | ✗ | 180375.06 | 13.99 | 0.00 | 75.48 | 69.23 | 64.14 | 69.23 |
| | OpenHands | ✓ | 328385.09 | 11.82 | 0.00 | 70.39 | 68.79 | 42.91 | 68.79 |
| | | ✗ | 514483.18 | 19.56 | 0.49 | 70.39 | 66.43 | 42.64 | 66.43 |
| | Moatless | ✓ | 163343.25 | 16.06 | 0.00 | 75.23 | 72.50 | 72.80 | 72.50 |
| | | ✗ | 285908.52 | 20.96 | 0.00 | 75.11 | 68.88 | 69.62 | 68.88 |

Table 2: Performance of different agents, with the same LM (DeepSeek-V3), on SWE-bench-Lite dataset. We organize metrics across 3 key dimensions: Tool Utilization, Logical Consistency and Efficiency. The best value is indicated with a green background, while the worst value is indicated with a orange background.

| Agent | Efficiency | | Logical Consistency | | | Tool Utilization | | Patch Correctness | |
|---|---|---|---|---|---|---|---|---|---|
| | # Tokens | # Turns | % Stuck-in-Loop | Inter-turns | Intra-turns | % Tool Suc. | Info-gain | % Apply | % Resolved |
| | | | SWE-bench-Lite | | | | | | |
| SWE-agent | 31732.79 | 5.00 | 0.00 | 69.39 | 59.84 | 52.42 | 59.84 | 70.23 | 2.68 |
| OpenHands | 470697.05 | 20.78 | 6.02 | 74.88 | 69.89 | 54.83 | 69.89 | 93.00 | 27.00 |
| Moatless | 317615.00 | 24.29 | 27.78 | 76.03 | 68.86 | 69.52 | 68.86 | 82.41 | 37.96 |
| | | | SWE-bench-Verified | | | | | | |
| SWE-agent | 154668.45 | 12.85 | 0.00 | 75.18 | 69.04 | 65.34 | 69.04 | 97.80 | 40.00 |
| OpenHands | 475116.28 | 17.92 | 0.39 | 70.39 | 67.40 | 42.70 | 67.40 | 93.85 | 22.31 |
| Moatless | 233043.83 | 18.85 | 0.00 | 75.16 | 70.44 | 70.99 | 70.44 | 86.84 | 40.00 |

tool calls contribute novel information. More details about the metric distribution between these two categories can be found in Appendix G.2.

> The comparison between resolved and unresolved trajectories reveals that successful resolution is strongly associated with trajectory-aware metrics. Hence, prioritizing these metrics is crucial for improving *% Resolved*.

## 4.2 RQ2: HOW EFFECTIVE DOES SWE-EVAL EXPOSE THE LIMITATIONS OF AGENTS?

To test SWE-eval's ability to expose failure modes of agents, we apply it to SWE-agent, OpenHands, and Moatless (Table 2). From SWE-eval's scores and rationales, we identify recurring failure modes. Appendix D provides details on agent evaluation.

**(1) SWE-agent performs worst because it optimizes for speed at the cost of depth.** It explores shallowly and uses the fewest turns and tokens (32k) yet achieves the lowest resolution rate, ranking last on *Inter-turns*, *Intra-turns*, *% Tool Suc.*, and *Info-gain*.

**(2) OpenHands underperforms because its dialogue policy is strictly linear, with no backtracking.** It cannot revisit earlier states when errors occur, so errors accumulate, especially after failed tool calls. Accordingly, it scores poorly on *Inter-turns* (74.88) and *% Tool Suc.* (54.83).

**(3) Moatless exhibits *Stuck-in-Loop* (27.78%) because it lacks an explicit loop-breaking mechanism.** Adding one would terminate unproductive rollouts and increase *% Resolved*. A second cause is its MCTS design: nodes with repeated responses are not recorded in history, so the LLM cannot observe redundancy or detect repetition.

To assess how well does SWE-eval expose LM shortcomings, we report four findings and their causes from Table 3. Further details of evaluating LMs are in Appendix E

Table 3: Performance of different LMs, with the same agent (SWE-agent), on SWE-bench-Lite.

| LM | Efficiency | | Logical Consistency | | | Tool Utilization | | Patch Correctness | |
|---|---|---|---|---|---|---|---|---|---|
| | # Tokens | # Turns | % Stuck-in-Loop | Inter-turns | Intra-turns | % Tool Suc. | Info-gain | % Apply | % Resolved |
| **SWE-bench-Lite** | | | | | | | | | |
| DeepSeek-V3 | 31732.79 | 5.00 | 0.00 | 69.39 | 59.84 | 52.42 | 59.84 | 71.67 | 2.73 |
| DeepSeek-R1 | 93568.82 | 14.70 | 0.34 | 78.76 | 64.23 | 81.21 | 64.23 | 80.61 | 23.47 |
| Qwen3-14B | 252911.48 | 17.54 | 48.99 | 76.46 | 61.23 | 27.95 | 61.23 | 22.15 | 0.00 |
| Qwen3-32B | 226342.03 | 15.93 | 41.22 | 75.06 | 58.67 | 28.05 | 58.67 | 23.31 | 0.00 |
| Mistral-small3.1-24B | 90785.23 | 6.49 | 0.00 | 71.56 | 52.59 | 31.84 | 52.59 | 22.22 | 0.00 |
| Devstral-24B | 201785.02 | 17.66 | 54.00 | 80.99 | 62.40 | 22.20 | 62.40 | 23.67 | 0.00 |
| Qwen2.5-Coder-7B | 153424.20 | 16.66 | 53.85 | 80.24 | 53.84 | 12.50 | 53.84 | 2.68 | 0.00 |
| Qwen2.5-Coder-14B | 176475.25 | 15.94 | 47.65 | 79.48 | 56.42 | 15.85 | 56.42 | 12.08 | 0.00 |
| Qwen2.5-Coder-32B | 206309.18 | 13.39 | 8.79 | 75.36 | 58.89 | 17.95 | 58.89 | 31.76 | 0.00 |

Table 4: Inter-group stability metrics among three groups (Prediction A, Prediction B, Human Annotation C). Confidence Interval (CI) establishes a numerical range, computed from sample data, to quantify the uncertainty of a statistical estimate.

| Metrics | Group | Value (95% CI) | | |
|---|---|---|---|---|
| | | Info-gain | Intra-turns | Inter-turns |
| Mean ± SD | Group A | 61.87 ± 15.79 | 60.19 ± 18.00 | 76.22 ± 10.50 |
| | Group B | 61.92 ± 15.40 | 60.26 ± 28.14 | 76.23 ± 10.71 |
| | Group C | 58.33 ± 22.30 | 55.48 ± 23.17 | 70.40 ± 22.79 |
| ICC(3,1) | Group A & B | 0.87 (0.86, 0.88) | 0.56 (0.53, 0.58) | 0.81 (0.80, 0.82) |
| ICC(3,k) | (A+B)/2 & C | 0.81 (0.80, 0.83) | 0.72 (0.70, 0.74) | 0.41 (0.36, 0.45) |
| Mean Diff | Group A & B | -0.05 (-0.33, 0.23) | -0.07 (-0.86, 0.71) | -0.01 (-0.24, 0.22) |
| | Group B & C | 3.54 (2.99, 4.09) | 4.71 (4.08, 5.35) | 5.81 (5.05, 6.58) |
| | Group A & C | 3.59 (3.04, 4.15) | 4.78 (3.80, 5.77) | 5.82 (5.05, 6.60) |

**(1) Agent-specific fine-tuning overfits tool schemas, weakening SWE-agent tool use.** Devstral fine-tuned on OpenHands scores 22.20 on *% Tool Suc.*, below Mistral-small-3.1 (31.84), indicating reduced cross-agent transfer.

**(2) Scaling LMs improves trajectories due to limited capacity to recover from tool errors.** In Qwen2.5-Coder, scaling from 7B to 32B lifts *% Apply* from 2.7% to 31.3%, with higher tool utilization and success (*Info-gain*: 16.5 to 36; *% Tool Suc.*: 12.5% to 18%).

**(3) On domain-specific tasks, scaling yields smaller gains for general models because domain data strengthens tool grounding in code models.** For Qwen2.5-Coder, scaling from 14B to 32B raises *% Apply* from 12% to 31.3%. For Qwen3, the same scaling increases it only from 22% to 23%. This gap aligns with stronger tool utilization in code models (*Info-gain*: 21.6 to 36).

**(4) LLMs outperform SLMs because scaling crosses a threshold that yields qualitative gains in tool use and Logical Consistency.** DeepSeek-R1 reaches 23% resolution, while all SLMs are 0%, consistent with higher *% Tool Suc.* and *Info-gain*.

> SWE-eval evaluates agents beyond patch correctness and explains why tasks succeed or fail. It diagnoses both agent policies and LMs via Tool Utilization and Logical Consistency, exposing concrete failure modes.

### 4.3 RQ3: How reliable and consistent is SWE-eval?

We employ three statistical metrics to examine the reliability and consistency of LLM-based evaluations by testing their stability and agreement with human annotations. To obtain high-quality human scores, we recruit three domain experts with substantial evaluation experience. Each expert independently scores the same trajectories using the same rubric as the LLM, and disagreements are resolved through iterative review. This procedure controls for rubric and data variation, isolating the evaluator effect. In our analysis, the two LLM-based predictions are denoted as groups A and B, while the human scores constitute group C.

**Evaluation Metrics** We use statistical reliability metrics to assess the soundness of SWE-eval. Suppose $N$ evaluation metrics are measured under a set of groups $G$, where $i = 1, \ldots, N$ indexes metrics and $g \in G$ indexes groups. Let $x_i^{(g)}$ denote the value of metric $i$ in group $g$. We report the following indicators: (1) Arithmetic Mean ± Standard Deviation (Mean ± SD). For each group, we summarize the distribution of metric values by reporting the sample mean and standard deviation. (2) Intraclass Correlation Coefficients (ICC). To quantify the stability of metric values across groups, we compute ICC(3, 1) and ICC(3, k) following the formulation of Shrout & Fleiss (1979); McGraw & Wong (1996); Koo & Li (2016); Liljequist et al. (2019). Here, metrics are treated as targets and groups as fixed raters, yielding a two-way mixed-effects model. Let $MS_{\text{ind}}$ denote the mean square between metrics and $MS_{\text{err}}$ the residual mean square from the two-way ANOVA (metrics × groups). The single-measure and average-measure forms are then defined as:

$$\text{ICC}(3, 1) = \frac{MS_{\text{ind}} - MS_{\text{err}}}{MS_{\text{ind}} + (k - 1)MS_{\text{err}}} \quad \text{ICC}(3, k) = \frac{MS_{\text{ind}} - MS_{\text{err}}}{MS_{\text{ind}}}$$

where $k$ denotes the number of groups (i.e., $|G|$). ICC(3, 1) measures the reliability of a single rater (e.g., multiple rounds of scoring by the same LLM), whereas ICC(3, k) captures the reliability of the aggregated mean rating across different raters (e.g., LLM and human experts). (3) Mean Difference (Mean Diff). We compute the bias between groups. The results are presented in Table 4.

**(1) Alignment of LLM Scores with Human Judgments.** We compute the ICC(3,k) between Group C and the average score of Groups A and B. LLM scores exhibit strong alignment with expert ratings on Info-gain (0.81), moderate alignment on Intra-turn (0.72), and weak alignment on Inter-turn (0.41). The mean differences between LLM groups and experts are +3.59 for Info-gain and +5.82 for Inter-turns between Group A and C, and +3.54 for Info-gain and +5.65 for Inter-turns between Group B and C.

**(2) The score distribution of LLM is reasonable.** Standard deviations across the three groups indicate that LLM ratings retain sufficient variance (SD $gt$ 10), comparable to human expert scores, ruling out score collapse and confirming that the ratings remain discriminative.

**(3) LLM scores are stable and consistent across multiple ratings.** The mean differences between LLM runs are minimal, with LLM consistently assigning higher scores than the experts (e.g., +3.59 in Info-gain, +5.82 in Inter-turns). The 95% confidence intervals exclude zero, indicating these differences are systematic rather than due to random variation. Additionally, the ICC(3,1) for Group A and B is 0.87 for Info-gain, 0.56 for Intra-turn, and 0.81 for Inter-turns, indicating strong stability and consistency in LLM ratings across multiple runs.

> LLM scores exhibit strong alignment with human ratings and a reasonable distribution. The scores are stable and consistent across multiple evaluations. These results confirm the reliability and effectiveness of SWE-eval.

## 4.4 CASE STUDY

SWE-eval exposes two critical shortcomings when Moatless addresses Django-12700. (1) Agents often become trapped in repetitive error loops (*Stuck-in-Loop*) due to **inadequate recovery mechanisms**. (2) Existing evaluation approach acquires excessively large patches (*# Line* $\geq$ 2k) by including extraneous **environment configuration files**.

Agents repeatedly commit the same error across multiple iterations, which is found by *Stuck-in-Loop*. This repetitive behavior is precisely quantified by continuous low scores: $\leq 45$ in *Intra-turns* and $\leq 40$ in *Info-gain*. These metrics specifically pinpoint the repeated rounds, such as rounds 20 through 30. Although LLM might acknowledge, "I keep repeating the same mistake.", the LLM continues to use invalid file editing parameters, perpetuating the loop. Although semantic errors are detected, the absence of robust recovery mechanisms allows initial errors to propagate unchecked. Consequently, the observed disconnect between error acknowledgment and corrective action highlights a fundamental gap between declarative understanding and procedural execution.

Our patch quality assessment revealed that generated patches exhibit excessive size (*# Line* $\geq$ 2k) and numerous code smells (e.g., over 100). This issue arises because current benchmarks often

fail to exclude common `.gitignore` entries during patch extraction. Consequently, irrelevant configuration directories, such as `.venv` and `.node_module`, are included. While these oversized patches might pass test cases, their bulk severely impedes accurate analysis of agent capabilities. If the patch fails, it is difficult to analyze where the patch is wrong, it may be an error in other files. This also significantly limits their real-world applicability. SWE-eval provides actionable insights for refining benchmark design.

## 5 RELATED WORK

**Coding Benchmarks** Code generation benchmarks have evolved from early single-file synthetic tasks, such as HumanEval (Yadav & Mondal, 2025) and MBPP (Austin et al., 2021), to comprehensive repository-level evaluation frameworks. A key milestone in this area is SWE-bench (Jimenez et al., 2023), which established an evaluation paradigm based on resolving real-world GitHub issues with verifiable code patches produced by coding agents. This paradigm has since been extended with multimodal contexts (Yang et al., 2024c), multilingual support (Zan et al., 2025). However, recent work argue that data in SWE-bench suffer from solution leakage and weak test cases(Aleithan et al., 2024). Improvements to SWE-bench address the static nature of the original benchmarks through dynamic issue curation (Zhang et al., 2025b) and enhanced unit test generation (Yu et al., 2025). TRAIL (Deshpande et al., 2025) articulates the need for robust and dynamic evaluation methods for agentic workflow traces, proposing a formal taxonomy of errors in agentic systems. Besides SWE-bench, Google's internal bug dataset GITS-Eval(Rondon et al., 2025) was curated, which further expanded the evaluation domain from open-source projects to industrial-scale enterprise projects. However, when evaluating the issue resolving abilities of coding agents, existing benchmarks ignore the multi-turn conversational trajectories inherent to agent-based issue resolution. Our SWE-eval addresses this gap by introducing trajectory-based multi-dimensional evaluation approach that evaluates both trajectories and patches.

**Coding Agent** Coding agents emerge, showcasing sophisticated issue resolution capabilities. Notable examples include SWE-Agent (Yang et al., 2024b), OpenHands (Wang et al., 2025), and Agentless (Xia et al., 2025). Moatless Tools integrates Monte Carlo Tree Search for self-improvement (aorwall, 2025), while SemAgent leverages semantic analysis to ensure patch completeness (Pabba et al., 2025). Furthermore, to better emulate human developer workflows, recent works have introduced multi-agent frameworks, where specialized agents in systems like HYPER-AGENT (Phan et al., 2024) and MAGIS (Tao et al., 2024) collaborate to handle complex tasks. Despite their advanced architectures, issue resolution remain outcome-centric, providing limited insight into the reasoning processes that lead to success or failure. SWE-eval pioneers trajectory-based evaluation, offering unprecedented visibility into an agent's decision-making processes while maintaining a rigorous assessment of final solutions.

## 6 DISCUSSION

The reliability of LLMs as automated code evaluators is a subject of debate. We demonstrate that the LLM-based evaluations are not arbitrary. They exhibit strong internal consistency and a systematic, predictable bias when compared to human ratings. This finding validates their use as a rational and scalable proxy for human evaluation.

Given limited compute, we did not use more expensive LLMs or very large datasets. To ensure reproducibility, we report empirical results on SWE-bench-Lite and SWE-bench-Verified in the main text and include the SWE-smith subset in the appendix.

## 7 CONCLUSION

We present SWE-eval, a trajectory-augmented evaluation framework with two layers: a trajectory layer that assesses efficiency, tool use, and logical consistency, and a patch layer that evaluates patch correctness. This separation exposes process vs. outcome errors, addressing the limitation of benchmarks that score only patches. In experiments, SWE-eval surfaces failures in agents and LMs and recommends targeted fixes.

## 8 ETHICS STATEMENT

Our work introduces SWE-eval, a multi-dimensional evaluation framework for assessing issue-resolving agents and language models in software engineering. While the framework itself is designed for analytical and diagnostic purposes, we recognize that automated code generation and patching carry potential risks, such as the introduction of insecure or non-robust code. To mitigate these concerns, we emphasize that SWE-eval is intended for research and development contexts to help improve the transparency and reliability of AI-assisted software engineering. Furthermore, all datasets and issue examples used in this study are derived from publicly available sources and contain no private or sensitive information. We adhere to responsible research norms and recommend that future applications of this framework prioritize fairness, accountability, and safety.

## 9 REPRODUCIBILITY

To ensure the reproducibility of our findings, detailed implementation instructions for SWE-eval can be found in Appendix C. Additionally, the source code is publicly accessible at `https://anonymous.4open.science/r/SWE-eval-73D0ED31`. These resources are intended to enable independent verification and replication of our results by the research community.

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

## A  APPENDIX INTRODUCTION

This appendix provides essential supplementary material, enhancing the main paper's findings and methodology. We first detail the **experimental setup**, ensuring complete transparency and reproducibility of our results. Subsequently, we address three core **research questions** (**RQ1, RQ2, RQ3**). To rigorously evaluate performance and ensure the validity of our findings, we include a **comparative analysis of metric values**, providing in-depth statistical breakdowns. A dedicated **case study** section then illustrates the practical application and effectiveness of our approach through concrete, real-world examples. Finally, we evaluate the **generalization** capabilities of the **SWE-eval** framework, demonstrating its robustness and broad applicability across diverse scenarios. This comprehensive documentation provides the empirical foundation necessary for thorough understanding, replication, and further research.

## B  USE OF LLMS

In this research, we utilized Large Language Models (LLMs) as an auxiliary tool to enhance research efficiency and the quality of the manuscript. We have strictly adhered to the principles of academic integrity, and the authors assume full responsibility for all final research content and conclusions. The specific use of LLMs is detailed below.

### LANGUAGE POLISHING AND COPY-EDITING

We employed an LLM as a general-purpose writing assistant to polish the language of our manuscript drafts. Its primary functions included:

- Correcting grammatical, spelling, and punctuation errors.
- Improving the clarity and fluency of the language.
- Refining sentence structures to better align with academic writing standards.

Throughout this process, all suggestions generated by the LLM were manually reviewed and confirmed by the authors to ensure they accurately conveyed our original intent and did not alter the core academic arguments of the paper.

### AUTOMATED EVALUATION OF CODING AGENT TRAJECTORIES

A primary contribution of this research is the investigation of using an LLM as a scalable, automated evaluator for coding agent trajectories. In this capacity, the LLM was not used as a simple assistant under human supervision, but rather as a core component of our evaluation methodology, whose performance and rationality were systematically analyzed.

The process was as follows:

- **Rubric-based Prompting:** We developed a detailed evaluation rubric with specific criteria for assessing trajectory quality, including aspects like information gain and turn-based dynamics. This rubric was then formulated into a precise prompt to guide the LLM.
- **Automated Scoring:** The LLM was used to autonomously score a large set of coding agent trajectories based on the provided prompt. This process was repeated to generate multiple sets of predictions for consistency analysis.
- **Rationality Analysis:** A significant part of our research, as detailed in Section 4.3, was dedicated to assessing the rationality of this LLM-based evaluation framework. We did not use the LLM's scores as ground truth directly. Instead, we rigorously compared the LLM's scores against each other and against a set of scores produced by human annotators. This analysis involved quantifying the LLM's performance in terms of statistical variance, inter-rater consistency (using Intra-class Correlation Coefficient), and systematic bias.

Our findings confirm that the LLM serves as a rational evaluation proxy: its judgments are stable, highly consistent, and exhibit a predictable, systematic difference from human ratings rather

| Size | Type | Model | #Param | Context Window |
|---|---|---|---|---|
| Large | General | DeepSeek-V3 | 671B | 64K |
| | Reason | DeepSeek-R1 | 671B | 64K |
| Small | General | Qwen3 | 14B | 40K |
| | | Qwen3 | 32b | 40K |
| | | Mistral-small3.1 | 24B | 128K |
| | Code | Qwen2.5-Coder | 7B | 32K |
| | | Qwen2.5-Coder | 14B | 32K |
| | | Qwen2.5-Coder | 32B | 32K |
| | SWE | Devstral | 24B | 128K |

Table 5: Studied LMs in this paper. LMs can be divided into two categories according to scale: LLMs and SLMs. SLMs has different scales. Model names include general, reason, code, and SWE-Fine-tuned. Context W. is Context Window.

than random noise. The use of the LLM in this manner was foundational to our research question concerning the validity of automated evaluation agents. The authors take full responsibility for the design of this methodology, the analysis of its results, and the conclusions drawn therefrom.

## C  EXPERIMENT SETUP

### C.1  STUDIED LMS

To substantiate the judiciousness of our Language Models (LMs) selection, Table 5 systematically presents the LMs chosen for this investigation, detailing their scale and architectural characteristics. This deliberate selection ensures a robust comparative analysis. By encompassing models across diverse scales and specialized domains, we validate the generalizability of our empirical findings.

Specifically, the **Large Language Models** (LLMs) category includes DeepSeek-V3 DeepSeek-AI (2025) and DeepSeek-R1 DeepSeek-AI et al. (2025), both featuring 671 billion parameters and a 64K context window. DeepSeek-V3 represents a state-of-the-art general-purpose model. DeepSeek-R1, with identical scale, is designed for reasoning tasks. This pairing enables a direct evaluation of general versus reasoning models.

To assess **Small Language Models** (SLMs), our study includes diverse models. Qwen3 Yang et al. (2025a) (14B, 32B parameters, 40K context) serves as a general-purpose LM, enabling investigation of general-domain model scale. For specialized applications, Qwen2.5-Coder Hui et al. (2024) (7B, 14B, 32B parameters, 32K context) is examined, providing insights into domain-specific performance scaling for code. Mistral-small3.1 Mistral (2025b) (24B parameters, 128K context) offers a general baseline. Devstral Mistral (2025a), fine-tuned from Mistral Small 3.1, excels at tool-use for codebase exploration and multi-file editing, powering software engineering agents. Its inclusion is crucial for evaluating domain-specific fine-tuning efficacy and practical performance in software engineering (*SWE*) applications.

To elucidate the factors governing language model efficacy, we systematically evaluated models **across varying scales and training paradigms** (domain-specific versus general-purpose). To assess performance scalability, we included models with diverse parameter counts (7B, 14B, 24B, 32B, 671B). This allowed us to identify the optimal balance between computational resources and achieved performance, guiding efficient deployment. Concurrently, we selected models across distinct types, *General*, *Reason*, *Code*, and *SWE-Fine-Tuned*, to robustly evaluate their strengths and weaknesses across varied linguistic and task domains. This comprehensive comparison provides critical insights into the training considerations necessary for developing effective LMs for specific applications.

**Prompt for Evaluating *Info-gain***

**Role positioning:** You are an expert judge of dialogue informativeness, with the ability to determine whether a statement meaningfully contributes new and relevant information.
**Task description:** Given **Current-sentence**, and **Previous-context**, your task is to evaluate the **informativeness** of **Current-sentence** - i.e., how much new, relevant, and helpful information it adds to the dialogue to solve the **Question**.
**Evaluation Dimensions (with weights):**
1.  **Novelty of Information (40%)** - Does the sentence introduce new facts, reasoning, explanations, or actions not already mentioned?
2.  **Topical Relevance (20%)** - Is the new information clearly connected to the main topic or current sub-task?
3.  **Utility to Dialogue Progression (20%)** - Does the sentence advance the conversation toward resolution, clarification, or decision-making?
4.  **Information Density (10%)** - Is the sentence compact yet informative, avoiding vague or redundant language?
5.  **Reasoning or Explanation Depth (10%)** - If reasoning is present, does it add meaningful insight or justification?
**Context:**
**Question**: {question}
**Current-sentence**: {cur}
**Previous-context**: {history_str}
**Scoring Instructions:**
-   Score **Current-sentence** from **0 to 100**, allowing any value in that range (e.g., 23, 62, 94).
-   Think about the rubric below before deciding your score.
**Important:**
-   Only output the final **int score**.
-   Do not output explanation, reasoning, or extra commentary.
-   The output must be a valid Python int.
Now provide the score below:

Figure 3: Prompt for Evaluating Info-gain

## C.2 FULL PROMPTS IN SWE-EVAL

For transparency and reproducibility, we fully disclose the prompt structure for the LM's dialogue trajectory evaluation within our SWE-eval. This structure, comprising three distinct templates (*Info-gain*, *Intra-turns*, *Inter-turns*), systematically controls variables to precisely evaluate prompt element influence on LM output. LM evaluation utilized compressed trajectories for efficient context provision. The prompt remained fixed across all instances for comparability, attributing LM output differences solely to the context.

(1) **Info-gain**. To precisely evaluate each tool invocation's utility within a query resolution process, we quantify its individual contribution. This quantification is essential for understanding each tool's efficacy. Information gain serves as a direct metric to assess the novel knowledge a tool invocation provides. A higher information gain value indicates a substantial contribution to query resolution, thereby validating the tool's effectiveness. Conversely, minimal information gain suggests redundancy or limited utility, enabling the identification of specific optimization areas. The prompt is shown in Figure 3.

(2) **Intra-turns** Reliable and coherent multi-turn interactions necessitate that each turn maintains consistency with all prior information, preventing contradictions. This comprehensive context includes the initial input $\hat{q}$ and all preceding premises $\text{RCU}_p^{(<i)}$. To quantify global inter-turn consistency, we assess contradictions between the current conclusion $\text{RCU}_c^{(i)}$ and the accumulated prior knowledge. Contradictions are primarily detected by analyzing associated tool execution results. This analysis provides concrete, verifiable evidence of inconsistency. The prompt is shown in Figure 4.

(3) **Inter-turns**. Evaluating an agent's ability to execute simulated user instructions within a turn requires assessing response rationality. A assessment of whether a LLM response resolves a user's instruction extends beyond mere logical consistency between the instruction and the immediate response. Because immediate consistency alone may not capture the full instruction intent. Crucially, subsequent user feedback provides vital context for determining true instruction resolution. This feedback reveals whether the instruction was genuinely fulfilled from the user's perspective. The prompt is shown in Figure 5.

---

**Prompt for Evaluating *Intra-turns***

**Role positioning:** You are a professional dialogue quality evaluator specializing in assessing the internal coherence and correctness of conversation summaries.

**Task description:** Given three single-sentence summary in interaction among **User-Instruction**, **LLM-Answer**, and **User-Feedback**. Now your task is to evaluate its **Intra-turnsness** - that is, whether the LLM-Answer logically follows from the User-Instruction. And whether this interaction is internally self-consistent, grammatically sound, and semantically coherent.

**Evaluation Criteria:**
1. **Topic Continuity (45%)**
   - Does the current sentence continue the core topic of the previous one?
2. **Logical Progression (20%)**
   - Is the current sentence a reasonable logical or conversational response to the previous one?
   - For example: a question is answered, a statement is elaborated, an action follows an intention, etc.
3. **Semantic Linkage (10%)**
   - Are there shared or related concepts/keywords?
   - Are references consistent via pronouns, synonyms, or hierarchical terms?
4. **Dialogue Act Alignment (10%)**
   - Do the speech acts (e.g., asking, suggesting, confirming, reacting) align in a coherent way?
   - Does the behavior fit typical dialogue flow?
5. **Temporal/Causal Fit (15%)**
   - Does the current sentence plausibly follow the previous one in time or causality?
   - Does it violate any commonsense expectations?

**Rules:**
- Only assess **coherence between User-Instruction and LLM-Answer** (not quality or informativeness).
- Do not consider whether the sentences are factually correct or grammatically sound.
- Output only the final **int score** (e.g., `49`). No explanations or comments.

**Context:**
User-Instruction: "{prev}"
LLM-Answer: "{cur}"
User-Feedback: "{next}"

**Scoring Instructions:**
- Score the sentence from **0 to 100**, allowing any value in that range (e.g., 23, 62, 94).
- Think about the rubric below before deciding your score.

**Important:**
- Only output the final **int score**.
- Do not provide explanation, analysis, or extra text.
- The output must be a valid Python int (e.g., 67)

Now provide the score below:

Figure 4: Prompt for Evaluating Inra-turns

| Agent | Tool Utilization | | Consistency | | Efficiency | | | | Patch Quality | | | |
|---|---|---|---|---|---|---|---|---|---|---|---|---|
| | Info-gain | % Tool Suc. | Intra-turns | Inter-turns | # In-tokens | # Out-tokens | Turns | % Stuck | # Lines | # Files | CC | # Smell |
| **All Instances (Unresolved+Resolved)** | | | | | | | | | | | | |
| DeepSeek-V3 | 51.2 | 52.4 | 40.8 | 36.8 | 31.7k | 1.9k | 5.0 | 0.0 | 15.8 | 3.0 | 1.2 | 0.5 |
| DeepSeek-R1 | 57.7 | 81.2 | 39.8 | 70.0 | 93.6k | 1.5k | 14.7 | 0.3 | 11.1 | 1.0 | 1.5 | 0.1 |
| Qwen3-14B | 10.6 | 27.9 | 48.5 | 51.7 | 252.9k | 18.5k | 17.5 | 49.0 | 5.0 | 0.5 | 0.3 | 0.6 |
| Qwen3-32B | 12.5 | 28.1 | 40.8 | 47.5 | 226.3k | 17.0k | 15.9 | 41.2 | 13.2 | 0.8 | 1.0 | 0.3 |
| Mistral-small3.1-24B | 42.4 | 31.8 | 32.2 | 39.5 | 90.8k | 17.2k | 6.5 | 0.0 | 12.1 | 0.7 | 0.9 | 0.3 |
| Devstral-24B | 35.9 | 22.2 | 46.9 | 66.9 | 201.8k | 12.5k | 17.7 | 54.0 | 11.1 | 0.4 | 1.1 | 0.3 |
| Qwen2.5-Coder-7B | 16.5 | 12.5 | 36.1 | 55.0 | 153.4k | 10.1k | 16.7 | 53.9 | 1.6 | 0.1 | 0.1 | 0.0 |
| Qwen2.5-Coder-14B | 21.6 | 15.8 | 39.6 | 55.5 | 176.5k | 11.9k | 15.9 | 47.6 | 4.6 | 0.3 | 0.4 | 0.1 |
| Qwen2.5-Coder-32B | 36.0 | 17.9 | 40.7 | 52.6 | 206.3k | 15.8k | 13.4 | 8.8 | 16.9 | 0.9 | 2.0 | 0.5 |
| **Unresolved Instances** | | | | | | | | | | | | |
| DeepSeek-V3 | 51.2 | 52.4 | 40.7 | 36.3 | 31.4k | 1.9k | 5.0 | 0.0 | 15.8 | 3.0 | 1.2 | 0.6 |
| DeepSeek-R1 | 56.0 | 83.2 | 39.1 | 70.4 | 107.9k | 1.7k | 16.3 | 0.4 | 9.0 | 0.9 | 1.6 | 0.1 |
| **Resolved Instances** | | | | | | | | | | | | |
| DeepSeek-V3 | 54.1 | 53.1 | 45.6 | 51.6 | 45.0k | 1.9k | 6.1 | 0.0 | 14.5 | 1.6 | 2.0 | 0.0 |
| DeepSeek-R1 | 63.1 | 74.6 | 42.1 | 68.7 | 46.9k | 810.7 | 9.5 | 0.0 | 18.1 | 1.2 | 1.1 | 0.1 |
| $AvgGain_{unresolved}^{resolved}$ | +9% | -5% | +10% | +20% | -7% | -26% | -9% | -50% | +47% | -9% | +22% | -44% |

Table 6: Performance of different LMs, with the same agent (SWE-agent), on SWE-bench-Lite dataset. *% Resolved* and *% Unresolved* are reported in parentheses. We organize metrics across four key dimensions: Tool Utilization, Logical Consistency, Efficiency and Patch Quality. We group instances by patch correctness: All, Unresolved, and Resolved. $AvgGain_{unresolved}^{resolved}$: average performance change of three agents from unresolved to resolved.

---

**Prompt for Evaluating *Inter-turns***

**Role positioning:** You are a professional evaluator of dialogue reasoning chains, with expertise in assessing the logical and semantic coherence between consecutive conversation turns.

**Task description:** Given two summaries of consecutive interaction rounds in a dialogue - **Current-sentence** and **Previous-sentence** - your task is to assess their **Inter-turnsness**, i.e., how well the current sentence logically and semantically follows from the previous one.

**Weighted Evaluation Criteria:**
- **Topic Continuity (45%)**: Does the current sentence maintain or meaningfully extend the topic from the previous one?
- **Intent Consistency (30%)**: Is the communicative intent (e.g., question-answer, elaboration, rebuttal) logically compatible across the two turns?
- **Reasoning Validity (15%)**: If implicit logic or inference is used, is it reasonable and well-grounded?
- **Other Pragmatic Coherence (10%)**: Includes reference clarity, tone matching, memory consistency, and turn appropriateness.

**You should penalize:**
- Abrupt or unjustified topic changes
- Logical contradictions or inferential gaps
- Inappropriate intent shifts (e.g., ignoring a question)
- Jarring tone or reference mismatch

**Context:**
**Question**: {question}
**Current-sentence**: {cur}
**Previous-context**: {history_str}

**Scoring Instructions:**
- Score **Current-sentence** from **0 to 100**, allowing any value in that range (e.g., 23, 62, 94).
- Think about the rubric below before deciding your score.

**Important:**
- Only output the final **int score**.
- Do not output explanation, reasoning, or extra commentary.
- The output must be a valid Python int.

Now provide the score below:

---

Figure 5: Prompt for Evaluating Inter-turns

# D    RQ1: EVALUATING AGENTS

## D.1    DETAILS OF EVALUATING AGENTS

To demonstrate the superiority of SWE-eval in evaluating agents, we details the performance of three agents (*SWE-agent* Yang et al. (2024a), *Openhands* Wang et al. (2025), and *Moatless* aorwall (2025)) on the SWE-bench-Lite dataset Jimenez et al. (2023), utilizing DeepSeek-V3. Table 2 in mainbody is organized by patch correctness, dividing the results into three sections: all instances (*unresolved* + *resolved*), *unresolved* instances, and *resolved* instances. The final row, $AvgGain_{unresolved}^{resolved}$, quantifies the average performance improvement when agents transit from an *unresolved* to a *resolved* state. The color intensity within the table indicates performance; darker shades represent superior results. For instance, a lower value for *# Turns* signifies better performance, while a higher value for *Info-gain* is desirable.

Our evaluation method is systematic. Metrics are organized into four key dimensions: Tool Utilization, Consistency, Efficiency, and Patch Quality. Tool Utilization measures *Info-gain %* and *Tool Suc.%*, reflecting tool application effectiveness. Consistency assesses *Intra-turns* and *Inter-turns* coherence. Efficiency includes *In-tokens*, *Out-tokens*, *# Turns*, and *# Stuck*, indicating resource consumption and operational smoothness. Patch Quality evaluates generated patches via *# Lines*, *# Files*, *CC*, and *# Smell*. Shading within the table signifies relative performance. Darker green generally denotes superior performance for beneficial metrics (e.g., higher percentages). Conversely, for metrics where lower values are preferable (e.g., token counts, *# Stuck*, *CC*, *# Smell*), darker green may indicate less favorable outcomes. The $AvgGain_{unresolved}^{resolved}$ row employs orange shading to highlight positive performance changes.

In general, the $AvgGain_{unresolved}^{resolved}$ statistical value quantifies the average performance improvement observed as instances transition from an unresolved to a resolved state across all three agents. Tool Utilization showed positive gains (*Info-gain* +14%, *Tool Suc.* +6%), signifying enhanced tool application. Consistency also improved (*Intra-turns* +12%, *Inter-turns* +19%), demonstrating more coherent agent behavior. Conversely, efficiency metrics reduced desirably (*In-tokens* -15%, *Out-tokens* -21%, *# Turns* -12%), indicating decreased resource consumption. A substantial *# Stuck*

decrease (-45%) further confirms that successful resolution profoundly reduces impasses. For Patch Quality, reductions in *# Lines* (-17%) and *# Files* (-8%) suggest more concise patches. Despite a slight *CC* increase (+16%) implying localized complexity, a significant *# Smell* decrease (-36%) confirms substantially cleaner resolved patches.

Analysis of the $AvgGain^{resolved}_{unresolved}$ metric reveals that **Inter-turns is the most significant factor** influencing *% Resolved*. This critical insight underpins the superior performance of Moatless compared to Openhands and SWE-agent. Moatless achieves this by organizing conversational nodes using a Monte Carlo tree approach, a strategic design choice that enables the provision of highly precise contextual and *Inter-turn* information. In contrast, Openhands and SWE-agent typically process all historical context indiscriminately, which can dilute the relevance of crucial *Inter-turn* dependencies and hinder optimal resolution.

**Across *All* instances**, distinct agents exhibited varied performance, a direct consequence of their differing design principles. Moatless consistently demonstrates superior Tool Utilization (65.3% *Info-gain*, 69.5% *Tool Suc.*) and Consistency (54.9% *Intra-turns*, 68.7% *Inter-turns*). This indicates its robust ability to leverage tools and maintain coherent interaction flows. However, Moatless also exhibits the highest Efficiency metrics, including *In-tokens* (317.6k), *Out-tokens* (4.8k), and *# Turns* (24.3), suggesting a more verbose and iterative problem-solving approach. Concurrently, it records the highest *# Stuck* rate (27.8%), indicating frequent impasses despite its verbosity. In Patch Quality, Moatless generates the most concise patches (7.8 *# Lines*, 0.9 *# Files*) but with higher complexity (*CC* = 2.1), implying greater complexity. Openhands shows moderate performance across most metrics and produces the cleanest patches (0.0 *# Smell*). SWE-agent exhibits the lowest Tool Utilization and Consistency. It is highly efficient, with minimal token usage and turns, and notably, never gets stuck (0.0% *# Stuck*). However, its patches are larger (15.8 *# Lines*, 3.0 *# Files*) and less clean (0.5 *# Smell*).

We performed more detailed comparative analysis **between *Resolved* and *Unresolved* instances**, for precisely identifying the factors governing the agent's issue resolution efficacy, . From *% Resolved*, Moatless demonstrates the highest resolution capability at 38%, followed by Openhands at 27%, while SWE-agent achieves only 3%. This disparity underscores substantial differences in their problem-solving efficacy. Analysis of resolved versus unresolved instances reveals consistent behavioral trends. Successful resolutions correlate with enhanced tool utilization and behavioral consistency. For example, Moatless's *Info-gain* % rises from 58.3% for unresolved cases to 76.7% for resolved ones, indicating that effective tool application and coherent agent actions are critical for successful problem resolution. Concurrently, efficiency metrics, including *In-tokens*, *Out-tokens*, and *# Turns*, generally decrease for resolved instances, suggesting that successful solutions typically require less computational effort and fewer interaction steps. A particularly critical finding is the substantial reduction in *# Stuck* rates for resolved instances; Moatless, for example, decreases from 37.3% to 12.2%. This demonstrates that successful resolution effectively mitigates impasses and improves task progression. Regarding patch quality, resolved patches are consistently more concise, evidenced by reductions in *# Lines* and *# Files*. Crucially, a significant decrease in *# Smell* counts for resolved patches indicates improved code cleanliness and quality upon successful completion.

# E  RQ2: EVALUATING LMS

## E.1  DETAILS OF EVALUATING LMS

To demonstrate the superiority of SWE-eval in evaluating LMs, Table 6 provides nuanced performance insights beyond simple success rates. Utilizing SWE-agent, it details trajectory- and patch-based metrics to analyze agent behavior and code quality across diverse categories, identifying specific LM strengths and weaknesses for software engineering tasks. The modular design measures tool interaction effectiveness, decision-making coherence, resource consumption, and generated code characteristics.

System performance is rigorously evaluated across five critical dimensions. *Tool Utilization* quantifies interaction efficacy by measuring *Info-gain* (information acquired) and *% Tool Suc.* (successful executions). These metrics are specifically chosen to assess effective tool application, paramount for successful problem-solving. *Logical Consistency* assesses decision-making coherence and behavioral robustness across turns. Employing *Intra-turns* for within-turn stability and *Inter-turns* for

cross-turn continuity, this approach verifies reliable system operation in dynamic, multi-step scenarios. *Efficiency* metrics provide insights into resource consumption and operational fluidity. *In-tokens* and *Out-tokens* track computational load; *Turns* denotes interaction length; and *% Stuck* identifies stagnation. These indicators are essential for pinpointing bottlenecks and optimizing practical deployment. Finally, *Patch Quality* characterizes generated code's maintainability and robustness. Assessed using *# Lines*, *# Files*, *Cyclomatic Complexity* (CC), and *# Smell*, these standard software engineering metrics comprehensively evaluate the long-term viability and ease of maintenance of the system's output.

The *Unresolved* instances and *Resolved* instances provide a direct comparison for DeepSeek-V3 and DeepSeek-R1. For DeepSeek-V3, resolving instances improves *Info-gain* from 51.2 to 54.1, and *Inter-turns* consistency from 36.3 to 51.6. DeepSeek-R1, when instances are resolved, shows a slight decrease in *% Tool Suc.* from 83.2 to 74.6 but a notable increase in *# Lines* from 9.0 to 18.1, alongside a reduction in *% Stuck* from 0.4% to 0.0%. The $AvgGain_{unresolved}^{resolved}$ row further elaborates on these trends, showing a +9% average gain in *Info-gain* and a substantial +20% gain in *Inter-turns* consistency for resolved instances. However, some efficiency metrics, such as *Out-tokens*, show a -26% average decrease, implying increased resource usage for resolved tasks. Patch quality metrics present a mixed picture, with *# Lines* increasing by +47% and *CC* by +22%, while *# Smell* decreases by -44%, suggesting that resolved patches are more extensive but exhibit fewer code quality issues.

Our analysis yields four principal conclusions::

**(1) Analysis of Parameter Increases on *% Tool Suc.* for Code-Specific vs. General Models** The data indicates that increasing parameter count yields more substantial improvements in *% Tool Suc.* for code-specific models compared to general models. For the general Qwen3 family, an increase from 14B to 32B parameters results in a marginal improvement in *% Tool Suc.* from 27.9% to 28.1%, a mere 0.2 percentage point gain. In contrast, the code-specific Qwen2.5-Coder family demonstrates more pronounced gains: the 7B model has a *% Tool Suc.* of 12.5%, which increases to 15.8% for the 14B model (a 3.3 percentage point increase), and further to 17.9% for the 32B model (an additional 2.1 percentage points, or 5.4 percentage points from 7B). This suggests that architectural specialization for code tasks allows for more effective leveraging of increased model capacity to enhance tool interaction success.

**(2) Comparison of Devstral and Mistral in *% Tool Suc.*** Contrary to the initial hypothesis, the fine-tuned Devstral-24B model does not marginally outperform its base model, Mistral-small3.1-24B, in *% Tool Suc.*. The data explicitly shows that Mistral-small3.1-24B achieves a *% Tool Suc.* of 31.8%, while Devstral-24B records a lower 22.2%. This indicates that, at least for this specific metric and dataset, the fine-tuning applied to Devstral-24B did not translate into improved tool success rates compared to the foundational Mistral model. This finding warrants further investigation into the fine-tuning methodology and its impact on specific performance dimensions.

**(3) Impact of Increasing Parameter Count on SLM Capabilities within the Same Architectural Family** Within the Qwen3 and Qwen2.5-Coder architectural families, increasing the parameter count does not consistently improve all SLM capabilities; performance varies across different metrics. For instance, in the Qwen2.5-Coder family, increasing from 7B to 32B parameters leads to significant improvements in *Info-gain* (16.5 to 36.0) and a drastic reduction in *% Stuck* (53.9% to 8.8%), indicating enhanced tool utilization and task completion reliability. However, this parameter increase also results in a degradation of efficiency metrics such as *In-tokens* (153.4k to 206.3k) and *Out-tokens* (10.1k to 15.8k), suggesting higher computational costs. Similarly, patch quality metrics like *# Lines* (1.6 to 16.9) and *# Smell* (0.0 to 0.5) worsen, implying that larger models might generate more verbose or less clean code despite improved task resolution. This nuanced outcome highlights that while scaling improves some core capabilities, it can introduce trade-offs in efficiency and direct output quality.

**(4) Comparison of LLMs and SLMs in Overall Task Resolution** Large Language Models (LLMs) generally demonstrate substantial outperformance over Small Language Models (SLMs) in overall task resolution, particularly in key indicators like *% Tool Suc.* and *% Stuck*. DeepSeek-R1, an LLM, achieves an impressive 81.2% *% Tool Suc.* and a near-perfect 0.3% *% Stuck* rate. Similarly, DeepSeek-V3 and Mistral-small3.1-24B exhibit excellent *% Stuck* rates of 0.0%. In contrast, most SLMs, such as Qwen3-14B and Qwen2.5-Coder-7B, show considerably lower *% Tool Suc.* (27.9%

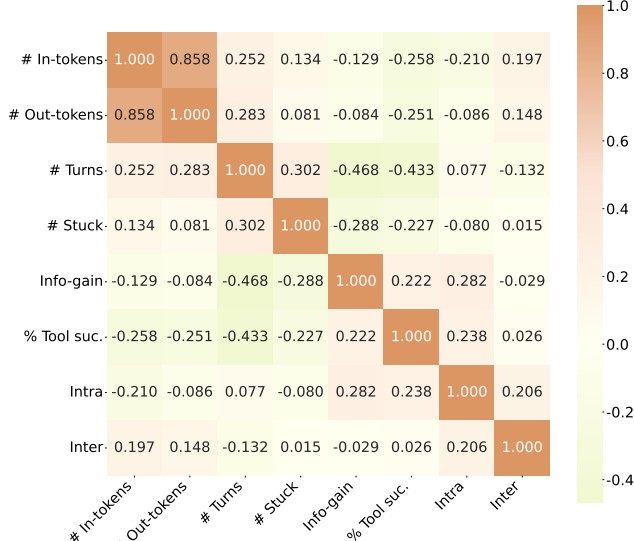

Figure 6: Spearman correlation heatmap of trajectory-based metrics, showcasing their independence.

and 12.5% respectively) and significantly higher *% Stuck* rates (49.0% and 53.9% respectively). While the Qwen2.5-Coder-32B model, a larger SLM, reduces its *% Stuck* to 8.8%, it still lags behind the top-performing LLMs in *% Tool Suc.* (17.9%). This disparity underscores the superior robustness and effectiveness of LLMs in completing complex tasks within this benchmark.

## F    RQ3: CORRELATION

### F.1    CORRELATION OF TRAJECTORY-BASED METRICS

The heatmap, presented as Figure 6, provides a Spearman correlation analysis of various trajectory-based metrics, primarily aiming to illustrate their independence. Each cell in the matrix represents the correlation coefficient between the metric on its row and the metric on its column. The color scheme of the heatmap is designed to intuitively convey both the strength and direction of these correlations. A divergent color gradient is employed, where shades of orange/brown indicate positive correlations, signifying that two metrics tend to increase or decrease together. Conversely, shades of green represent negative correlations, meaning that as one metric increases, the other tends to decrease. The intensity of the color directly corresponds to the absolute magnitude of the Spearman coefficient: darker hues denote stronger correlations (values closer to 1 or -1), while lighter shades or white indicate weaker or negligible correlations (values closer to 0). This visual encoding allows for rapid identification of the most significant relationships within the dataset.

The color scheme of the heatmap visually encodes the strength and direction of these correlations. A gradient from light green/yellow to dark orange/brown is employed. Darker orange/brown hues indicate stronger positive correlations (approaching +1.0), signifying that as one metric increases, the other tends to increase proportionally. Conversely, lighter green/yellow shades denote negative correlations (approaching -1.0), meaning that as one metric increases, the other tends to decrease. Colors closer to the center of the spectrum (lighter yellow/white) represent correlations near zero, indicating a weak or negligible linear relationship between the metrics. The diagonal, which correlates each metric with itself, is consistently 1.000 and depicted in the darkest orange, as expected.

The heatmap analysis reveals that several key metrics exhibit weak correlations, **underscoring their distinct contributions to system evaluation**. For instance, a moderate negative correlation ($r = -0.468$) exists between *Info-gain*, *Intra-turns*, and *Inter-turns*. This relationship indicates that higher information gain is associated with fewer conversational turns, suggesting an efficient interaction where users achieve their goals with less verbose dialogue. Conversely, the correlation between *#*

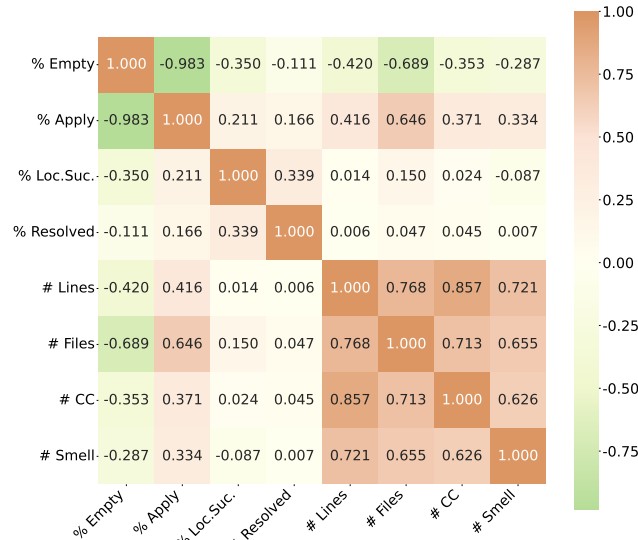

Figure 7: Spirman coefficient of patch-based metrics correlation heatmap.

*Stuck* and *Inter-turns* is nearly negligible ($r = 0.015$), demonstrating that the frequency of the system becoming stuck has almost no linear relationship with the overall number of interactions. Similarly, the weak negative correlation ($r = -0.081$) between *Info-gain* and *# Stuck* implies that system impasses have minimal impact on the amount of information conveyed. These consistently weak correlations across diverse metric pairs are not merely observations; they are a deliberate design outcome, chosen to ensure each metric captures a unique facet of system performance. This independence is paramount for comprehensive system evaluation because it prevents redundancy among measures, allowing researchers to gain a granular and nuanced understanding of specific strengths and weaknesses without confounding factors. By isolating distinct aspects of interaction quality, these metrics collectively provide a more complete and interpretable profile of system behavior.

## F.2 CORRELATION OF PATCH-BASED METRICS

To systematically analyze the interdependencies among our proposed patch metrics, we conduct a correlation analysis, visually represented in Figure 7. This figure presents Spearman's rank correlation coefficients, providing crucial insight into the intrinsic relationships among our patch-based metrics. This visualization is constructed to identify potential redundancies and reveal underlying dependencies, which are vital for informing subsequent *feature selection* processes in our analysis of software patches. By quantifying the *monotonic association* between each pair of metrics, we gain critical insights into how different patch characteristics co-vary. This understanding is crucial for building robust predictive models and deriving meaningful conclusions regarding patch quality and impact. The color scheme aligns with that used in the *Trajectory-based heatmap* for visual consistency. The color scale indicates correlation strength and direction, with lighter shades signifying weaker relationships.

Specifically, **we systematically quantify the relationships among diverse patch metrics** through correlation coefficients to establish the empirical rigor of the SWE-eval design. Our analysis reveals that several metric pairs exhibit low correlation, approaching zero, indicating they capture distinct, orthogonal patch aspects. For example, *% Resolved* correlates negligibly with *# Lines* (0.006) and *# Smell* (0.007), and *% Loc.Suc.* weakly correlates with *# Lines* (0.014) and *# Smell* (-0.087). These low correlations are critical: they confirm unique information contributions from each metric, enabling comprehensive patch characterization and mitigating multicollinearity. Conversely, strong positive and negative correlations, visualized by dark orange and green hues, highlight significant interdependencies. A prominent example is the strong negative correlation of -0.983 between *% Empty* and *% Apply*, signifying highly redundant information. Furthermore, patch size and complexity metrics, such as *# Lines*, correlate strongly with *# Files* (0.768), *# CC* (0.857), and *# Smell*

| Agent | Tool Utilization | | Consistency | | Efficiency | | | | Patch Quality | | | |
|---|---|---|---|---|---|---|---|---|---|---|---|---|
| | Info-gain | % Tool Suc. | Intra-turns | Inter-turns | # In-tokens | # Out-tokens | Turns | % Stuck | # Lines | # Files | CC | # Smell |
| All Instances (Unresolved+Resolved) | | | | | | | | | | | | |
| SWE-agent | 53.6 | 52.8 | 42.5 | 37.8 | 35.5k | 1.9k | 5.3 | 0.0 | 19.6 | 4.2 | 1.3 | 0.6 |
| Openhands | 63.8 | 54.6 | 34.9 | 69.5 | 489.7k | 4.7k | 12.2 | 3.0 | 27.9 | 1.5 | 4.1 | 0.8 |
| Moatless | 64.3 | 66.5 | 55.8 | 70.9 | 371.9k | 5.5k | 29.0 | 41.7 | 6.9 | 0.8 | 1.8 | 0.0 |
| Unresolved Instances | | | | | | | | | | | | |
| SWE-agent | 53.8 | 52.3 | 42.5 | 37.2 | 35.4k | 1.9k | 5.3 | 0.0 | 19.9 | 4.2 | 1.3 | 0.6 |
| Openhands | 62.3 | 52.5 | 34.4 | 68.2 | 569.5k | 5.1k | 12.2 | 1.5 | 32.4 | 1.6 | 4.3 | 0.9 |
| Moatless | 58.2 | 62.2 | 54.3 | 69.2 | 433.0k | 5.8k | 32.9 | 53.9 | 6.5 | 0.7 | 1.9 | 0.0 |
| Resolved Instances | | | | | | | | | | | | |
| SWE-agent | 46.8 | 73.2 | 45.0 | 66.4 | 38.5k | 1.9k | 5.5 | 0.0 | 8.0 | 1.0 | 1.5 | 0.0 |
| Openhands | 67.0 | 59.1 | 36.0 | 72.3 | 320.1k | 3.8k | 12.1 | 6.2 | 18.5 | 1.3 | 3.7 | 0.6 |
| Moatless | 80.4 | 77.6 | 59.6 | 75.2 | 213.0k | 4.7k | 18.8 | 10.0 | 7.9 | 1.1 | 1.8 | 0.0 |
| $AvgGain^{resolved}_{unresolved}$ | +11% | +26% | +7% | +31% | -29% | -15% | -13% | +81% | -27% | -14% | -1% | -45% |

Table 7: In instances generated by SWE-smith, we report performance of SWE-agent and Openhands with Deepseek-V3. Proof SWE-eval can be applied to more than just SWE-bench-Lite. It can also be applied to other benchmarks similar to SWE-Bench. We organize metrics across four key dimensions: Tool Utilization, Logical Consistency, Efficiency and Patch Quality. We group instances by patch correctness: All, Unresolved, and Resolved. $AvgGain^{resolved}_{unresolved}$: average performance change of three agents from unresolved to resolved.

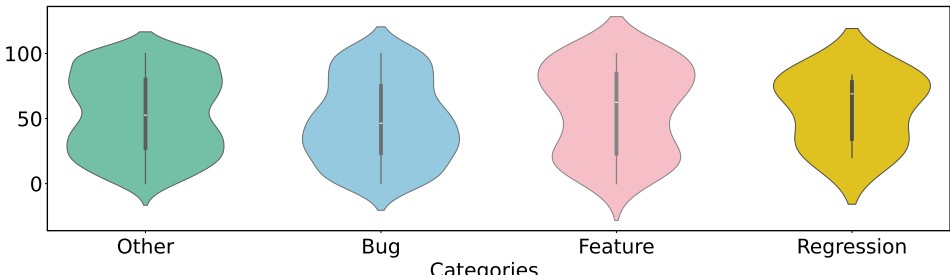

Figure 8: Violin plots of *% Tool Suc.* across different categories of issues, with SWE-agent and DeepSeek-R1

(0.721). This suggests larger patches inherently possess higher cyclomatic complexity and more code smells. Identifying such strong correlations is paramount for effective feature engineering, guiding metric prioritization or combination to enhance patch quality by systematically reducing redundancy.

# G COMPARATIVE ANALYSIS OF METRIC VALUE

## G.1 DIFFERENCES ACROSS ISSUE CATEGORIES

Figure 8 is integral to empirically substantiate the performance characteristics of the proposed tool across distinct **issue categories**. By employing violin plots, we move beyond mere central tendency, offering a comprehensive depiction of the *% Tool Suc.* distribution for each category. This approach allows for a nuanced understanding of where the tool performs consistently well, where it struggles, and the overall variability of its accuracy, thereby providing deeper insights into its robustness and applicability.

**By analyzing *% Tool Suc.* across various issue types**, we quantified their influence on the agents. For all categories, the median *% Tool Suc.*, indicated by the white dot within each violin, consistently hovers around 50. This suggests a general baseline performance across diverse issue types. However, the shapes of the violin plots, which represent the probability density of the *% Tool Suc.* data, illustrate significant distributional differences. The Other category exhibits a broader distribution, with notable density at both the median and near 100, implying that while some "other" issues yield moderate accuracy, a substantial portion achieves very high accuracy. In contrast, both Bug and Regression categories show a more pronounced density towards the lower end of the *% Tool Suc.* spectrum, specifically around 0, alongside the median concentration. This indicates that the tool

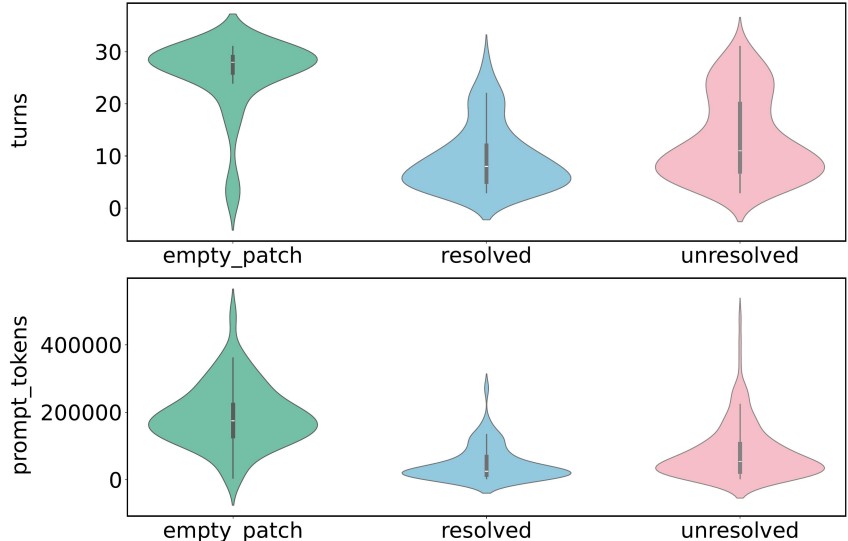

Figure 9: Violin plots of turns, In-tokens and completion_tokens across three kinds of patches (empty_patch, *Resolved* and *Unresolved*) with SWE-agent + Deepseek-r1

frequently encounters instances of very low accuracy when addressing Bug and Regression issues, suggesting these areas might represent specific challenges for the tool's current design. The Feature category, while also centering around a median of 50, displays a relatively uniform distribution across the accuracy range, implying a more varied performance profile without strong concentrations at either extreme, though a density near 0 is still present. These detailed distributional insights are crucial for identifying specific areas for targeted improvements, such as tool refinement.

### G.2 Differences Based on Patch Correctness

To characterize the numerical distribution of key performance indicators across diverse instances within the SWE-eval, we visualized all data points using violin plots in Figure 9. This figure provides critical insights into the 'SWE-agent + Deepseek-r1' system's operational efficiency by illustrating the distributions of *turns* and *In-tokens* across three distinct patch outcomes: *empty_patch*, *resolved*, and *unresolved*. We selected these metrics as they fundamentally indicate interaction depth and computational cost, respectively. Analyzing their distributions identifies scenarios of optimal and suboptimal agent efficiency.

The violin plots reveal distinct patterns in the agent's interaction depth, measured by *# Turns*. *Empty_patch* outcomes show the highest median *# Turns* and widest distribution, indicating extensive, prolonged exploration even when no code modifications are required. This inefficiency highlights a critical area for improvement: the agent's ability to quickly ascertain the absence of necessary changes to reduce operational time. Conversely, *resolved* tasks exhibit significantly lower median and tighter *# Turns* distribution. This demonstrates successful resolutions are achieved through a more focused, efficient interaction sequence, optimizing the agent's operational flow. *Unresolved* tasks, while having a median *# Turns* count comparable to *resolved* tasks, display a broader upper tail. This suggests some failures manifest only after substantial interactions, underscoring the significant cost of unproductive attempts.

Analysis of *In-tokens*, quantifying total tokens sent to the language model, reinforces these observations regarding resource expenditure. The *empty_patch* category consumes the most *In-tokens* by a substantial margin, with a very broad distribution. This elevated token usage directly correlates with high *turn* counts, as each interaction necessitates sending prompts. This finding underscores a significant inefficiency: considerable computational resources are expended on tasks yielding no functional patch. In contrast, *resolved* tasks demonstrate the highest token efficiency, displaying a much lower median and tighter *In-tokens* distribution, consistent with reduced *turn* counts. This efficiency for successful outcomes indicates optimized communication with the language model. For

*unresolved* tasks, the *In-tokens* distribution is generally higher than for *resolved* tasks, particularly in its upper range. This emphasizes that failed attempts incur substantial token costs, necessitating strategies to detect and gracefully exit unproductive paths earlier. Collectively, these distributions provide a robust empirical basis for understanding the agent's resource footprint and pinpointing specific performance enhancements.

# H  GENERALIZATION: SUPPLEMENT THE EVALUATION OF DIVERSE BENCHMARKS WITH SWE-EVAL

The SWE-eval framework adapts diverse benchmark, provided they conform to the SWE-bench trajectory and patch format. This standardization is essential for seamless integration and consistent processing. We utilized SWE-smith Yang et al. (2025b), a novel pipeline that synthesizes test-breaking task instances from Python codebases, to generate training data. To ensure the quality and relevance of our evaluation, we developed specific filtering rules for the raw SWE-smith output.

To ensure the quality, relevance, and solvability of instances within the dataset created by SWE-smith, we propose two rigorous filtering criteria. **(1) Complexity filtering** is a primary consideration. We exclude trivial instances involving minimal code changes (e.g., 1-2 lines) because they offer limited challenge for resolution systems. Instead, we retain instances necessitating modifications across multiple files or functions, which better reflect real-world software engineering complexity and demand more sophisticated problem-solving capabilities. **(2) Quality assurance** measures are applied through stringent content and environment checks. Problem description clarity is a key quality assurance measure. We artificially exclude instances with ambiguous descriptions or insufficient reproduction steps, as these hinder accurate problem comprehension and solution development. Only instances providing clear gold patch and precise error descriptions are retained, ensuring that the problem statement is unambiguous and actionable.

These rules were designed to select complex, solvable instances, yielding 976 refined samples. Considering computational cost and efficiency, we evaluated SWE-eval's performance on this curated subset using DeepSeek-V3 and SWE-agent. This focused evaluation demonstrates SWE-eval's extensibility to other benchmarks. The capacity for expansion relies on architectural compatibility and robust performance on representative samples, not merely dataset volume.

Performance metrics are detailed in Table 7. Table 7 provides empirical evaluation of SWE-agent, Openhands, and Moatless across Tool Utilization, Consistency, Efficiency, and Patch Quality, distinguishing between resolved and unresolved instances to illuminate agent capabilities and guide future design. This systematic comparison is crucial for understanding their operational profiles.

Our comparative evaluation reveals distinct performance profiles for Moatless, SWE-agent, and Openhands across key metrics, highlighting their specialized strengths in tool utilization, operational efficiency, and patch quality. Moatless generally excels in Tool Utilization, as evidenced by its highest *Info-gain* (64.3) and *% Tool Suc.* (66.5), and demonstrates superior Consistency in both *Intra-turns* (55.8) and *Inter-turns* (70.9), indicating effective tool leverage and coherent interactions. Conversely, SWE-agent consistently achieves superior Efficiency, consuming significantly fewer *In-tokens* (35.5k), *Out-tokens* (1.9k), and *Turns* (5.3), and notably maintaining a 0.0 *% Stuck* rate across all and unresolved instances. Regarding Patch Quality, SWE-agent produces cleaner, more focused code changes with the lowest *# Lines* (19.6 overall, 8.0 resolved), *# Files*, *CC*, and 0.0 *# Smell* in resolved instances. Openhands offers a balanced performance profile, often falling between the two.

To understand the critical differences between resolved and unresolved instances, we analyze the *AvgGain* row, revealing key characteristics of successful problem resolution. The *AvgGain* row, comparing resolved to unresolved instances, reveals that successful resolutions correlate with improved Tool Utilization (+11% *Info-gain*), Consistency (+31% *Inter-turns*), and Efficiency (e.g., -29% *In-tokens*, -13% *Turns*), alongside better Patch Quality (-45% *# Smell*). However, a notable +81% increase in *% Stuck* for resolved instances suggests that achieving resolution might involve navigating more complex or challenging scenarios, leading to a higher propensity for encountering temporary impasses. This analysis highlights distinct agent strengths and the inherent complexity of successful problem resolution, informing the development of more robust automated systems.

