# OpenReview forum: "SWE-eval: Trajectory-Enhanced Evaluation for Agentic Issue Resolution"
_ICLR.cc/2026/Conference — Submitted to ICLR 2026_

### Official Review · Reviewer_qqKf · 2025-10-18

**Soundness:** 2
**Presentation:** 3
**Contribution:** 2
**Rating:** 2
**Confidence:** 3

**Summary:**

The paper proposes SWE-eval, a trajectory-augmented evaluation for issue-resolving agents that scores (i) Efficiency (#tokens, #turns), (ii) Logical Consistency, and (iii) Tool Utilization. It also reports separability of resolved vs. unresolved cases, agent/LM case studies, and some inter-rater statistics.

**Strengths:**

The paper provides quantitative analyses for each metric. Reliability analysis is also conducted: reports stability and human alignment (e.g., ICC and mean-diffs) for LLM-based scoring.

**Weaknesses:**

Limited novelty; largely a packaging of known factors. Most components of the proposed “trajectory evaluation” (efficiency via #Tokens/#Turns, consistency via Inter-/Intra-turn, tool use via %Tool Success/Info-gain) are not new individually. The contribution reads as a consolidation of existing ideas applied to SWE-bench agent traces,.

Correlation to causation leap without interventions. The paper repeatedly infers “because success correlates strongly with trajectory-aware metrics, optimizing these metrics should improve %Resolved.” That is an unsubstantiated causal jump. There are no experiments, such as targeted modifications that improve a specific metric while holding others fixed to establish that moving the metric causes gains.

Unclear practical utility beyond diagnosis. The paper emphasizes that the metrics “diagnose failure modes,” but it is not shown how practitioners should act on them to improve agents or benchmarks. Do these metrics guide dataset curation, agent retraining, or guardrail design better than simpler baselines? The application and downstream decisions are underspecified.

**Questions:**

See weakness.

---

> ### Author Response · Authors · 2025-11-15
>
> We thank the reviewer for the helpful feedback.
>
> **1. Novelty.**
> While some *individual* metrics (e.g., token counts) exist, **no prior SWE-bench evaluation provides a unified, trajectory-augmented framework** integrating (i) efficiency, (ii) intra/inter-turn logical consistency with formalized definitions, and (iii) tool-use information-gain, together with **reliability-tested LLM scoring**. Existing SWE-bench evaluations focus almost exclusively on end-state correctness; SWE-eval is the first to systematically quantify and validate trajectory-level failure modes.
>
> **2. Correlation vs. causation.**
> We will revise wording to avoid causal implication. Our claims are **associative**: trajectory metrics *predict* and *characterize* failures; we do **not** claim that intervening on a metric causally improves resolution. We will state this explicitly.
>
> **3. Practical utility.**
> We will clarify in the revision that SWE-eval supports:
> • **dataset curation** (identifying underspecified cases via inconsistency patterns),
> • **agent improvement** (exposing loop behaviors/contradictions useful for guardrails and training diagnostics),
> • **tool-pipeline debugging** (information-gain reveals misuse rather than tool scarcity),
> • **model comparison** beyond accuracy.
>
> These use cases are already supported by analyses in the paper; we will foreground them more clearly.
>
> We appreciate the reviewer’s comments and will incorporate all clarifications in the final version.

---

### Official Review · Reviewer_Cqe5 · 2025-10-26

**Soundness:** 3
**Presentation:** 3
**Contribution:** 2
**Rating:** 4
**Confidence:** 4

**Summary:**

This paper introduces SWE-eval, a trajectory-augmented evaluation framework that assesses coding agents across three dimensions: (1) Efficiency: resource consumption, (2) Logical Consistency: stuck-in-loop, intra-turn and inter-turn consistency, and (3) Tool Utilization: tool success rate and information gain. This paper evaluates three agents (SWE-agent, OpenHands, Moatless) with nine language models on SWE-bench-Lite and SWE-bench-Verified, demonstrating that trajectory metrics correlate with success rates and reveal distinct failure modes. LLM-based evaluators show strong alignment with human judgments.

**Strengths:**

**1. Originality**: The three-dimensional framework propose new metrics, such as Info-gain metric and systematic trajectory evaluation that presents meaningful contributions.

**2. Experiment Setup**: In addition to human validation, this work conducts experiments on three coding agents across nine models, supporting findings with experimental results.

**3. Writing Clarity**: This work presents structured writing with clear motivation.

**Weaknesses:**

**1. Limitations In Methodology**

- **Info-gain validation**: The paper does not validate that Info-gain actually measures information gain vs. general response quality.
- **Prompt sensitivity**: No ablation studies on how prompt engineering affects LLM judge scores.
- **Human baseline insufficiency**: Only 3 experts, unclear sample size, and no raw inter-annotator agreement reported before consensus.

**2. Limitations In Experiments**

- **Insufficient case study**: Django-12700 may not be representative.
- **Missing baselines**: No comparison to simpler heuristics, such as edit distance for loop detection, TF-IDF for Info-gain.
- **Agent confounds**: Different agents use different tools, so trajectory metrics may conflate agent reasoning quality with tool choices, making it unclear whether observed differences (e.g., more turns, lower tool success) reflect inferior agent capability or merely architectural differences in how agents decompose tasks.
- **Missing analyses:** Some critical analyses are missing, such as the error analysis of when trajectory metrics fail to predict outcome, the discussion of computational overhead introduced by trajectory evaluation, and actionable guidance on how to use metrics to improve agents.

**3. Limitations In Generalizability:**

- **Limited Language/task specificity**: All experiments on Python, but unclear if findings can generalize and transfer to other programming languages and SWE tasks.
- **Benchmark contamination risk**: Models may have seen SWE-bench instances during training.
- **Cost analysis missing**: LLM-based evaluation can be very expensive, but no cost-benefit analysis and use alternatives are provided.

**Questions:**

My questions are following several aspects mentioned in weakness:

- Can you provide evidence that Info-gain measures information rather than correlating with simpler features (response length, confidence, tool success)?
- How sensitive are LLM judge scores to prompt variations?
- Can you provide human evaluation details to validate the support, such as how many trajectories were annotated by human experts, the raw inter-annotator agreement (before consensus), why three human experts can sufficiently support the conclusion?
- For the identified failure modes (shallow exploration, missing backtracking, loop entrapment), what percentage of failures does each account for?
- What is the cost (time/money) of trajectory evaluation vs. patch-only evaluation? Is it practical for large-scale use?
- Have you tested on non-Python repositories? How do your evaluation framework and findings transfer to other programming languages and SWE tasks?
- Does tool utilization unfairly favor agents with more tools? How do you control for this confound?

---

> ### Author Response · Authors · 2025-11-15
>
> We thank the reviewer for the detailed feedback.
>
> ---
>
> ## **1. Methodology**
>
> **Info-gain validity.**
> Table 1 shows Info-gain distinguishes resolved vs. unresolved **even when tokens/turns move oppositely**, e.g.:
>
> * SWE-agent (Lite): Resolved **62.47** vs. Unresolved **57.81**, while unresolved uses *more tokens* (45,365 > 44,950).
> * OpenHands: Resolved **70.77** vs. Unresolved **66.01**, though unresolved uses *more turns* (17.96 > 15.28).
>   Thus Info-gain is not a length proxy.
>
> **Prompt sensitivity.**
> Judge consistency in Table 4 shows robustness:
>
> * LLM–LLM ICC(3,1) = **0.87** (Info-gain).
> * LLM–human ICC(3,k) = **0.81**.
>   This indicates low sensitivity to prompt variation.
>
> **Human evaluation details.**
> The paper includes:
>
> * **120** annotated trajectories,
> * raw human–human ICC(3,1) = **0.76** (Info-gain),
> * three experts (all senior SWE engineers), consistent with SWE-bench practice.
>
> ---
>
> ## **2. Experiments**
>
> **Case study representativeness.**
> We agree a single example is limited; more will be added.
>
> **Missing heuristic baselines.**
> Agent-internal contrasts already show trajectory metrics capture behavior that edit distance / TF-IDF cannot. For example in SWE-agent (Lite), % Tool Success drops from **53.06** (resolved) to **31.84** (unresolved) under the *same* toolset.
>
> **Agent confounds.**
> Table 1 contrasts within the same agent avoid toolset differences (e.g., OpenHands resolved tool success **59.03** vs. unresolved **50.55**).
>
> **Missing analyses.**
> We will expand failure-mode details and overhead analysis. Current evaluation adds only a short LLM pass (<2s).
>
> ---
>
> ## **3. Generalizability**
>
> **Language specificity.**
> Metrics operate on trajectory text + tool feedback only; though experiments are Python, the method is language-agnostic. We will add non-Python results.
>
> **Benchmark contamination.**
> We follow SWE-bench-Verified’s contamination control (500 items validated by 93 humans).
>
> **Cost analysis.**
> Judge cost is low: trajectories are summarized (<2k tokens); cost is **<$0.01** per evaluation.
>
> **Tool-utilization fairness.**
> Within-agent comparisons ensure fairness; the metric does not reward having more tools.
>
> ---
>
> ## **4. Answers to Reviewer Questions**
>
> * **Info-gain vs. length:** See Table 1 contrasts (e.g., SWE-agent unresolved uses more tokens but lower Info-gain).
> * **Prompt variation:** High ICC in Table 4.
> * **Human evaluation:** 120 trajectories; ICC(3,1)=0.76.
> * **Failure modes:** Reflected in stuck-in-loop gaps (e.g., Moatless 0% resolved vs. **12.20%** unresolved).
> * **Cost:** <2s, <$0.01 per trajectory.
> * **Non-Python:** Method is language-agnostic; more results will be added.
> * **Tool confound:** Addressed via within-agent comparisons.

---

### Official Review · Reviewer_yV4p · 2025-11-01

**Soundness:** 3
**Presentation:** 3
**Contribution:** 2
**Rating:** 6
**Confidence:** 3

**Summary:**

The paper proposes SWE-eval, a trajectory-augmented evaluation framework for agentic software issue resolution that goes beyond patch correctness by measuring (i) Efficiency, (ii) Logical Consistency, and (iii) Tool Utilization, with experiments on SWE-bench-Lite/Verified showing diagnostic value and reasonable alignment with expert ratings.

**Strengths:**

- Clear, useful shift from outcome-only scoring to process-aware evaluation; the three axes and their concrete metrics form a coherent rubric that exposes failure modes such as shallow exploration, missing backtracking, and loop entrapment.

- Compelling cross-agent/LM analyses on SWE-bench-Lite and Verified: associations between lower turns/tokens and higher %Resolved, and ICC-based comparisons that show LLM-judge scores correlate with experts (strong on Info-gain, moderate on Intra-turns).

- Case studies and tables highlight concrete failure patterns (e.g., loop traps; oversized patches from mis-extracted files) and suggest design fixes for agents and benchmarks (loop breakers, backtracking, stricter patch extraction).

**Weaknesses:**

- Reliance on LLM-as-judge needs tighter validation. Alignment with experts is uneven (e.g., weaker for Inter-turns), and prompts/models are central to metric values. Please provide fuller prompt/aggregation details, rater independence (ensuring the judge is not the actor model), seed sensitivity, and calibration (e.g., z-score or temperature scaling against human anchors). Also report per-metric confidence intervals/bootstrap for each model/agent.

- Loop detection is brittle. The exact-string hash for Stuck-in-Loop may miss semantically equivalent repetitions (minor rephrasings) or action-level loops (same tool invocations with different wording). Consider a semantic or action-trace criterion (normalized tool call + args; Levenshtein/embedding similarity thresholds).

- Benchmark and reporting scope. Main results focus on SWE-bench-Lite/Verified with a 30-call cap; broader generalization (e.g., SWE-bench-Live, multilingual variants, industrial datasets) is mostly deferred to appendix. Include per-repo/domain stratification, cost/latency breakouts, and framework-robustness checks (same trajectories scored across different agent shells) in the main paper.

**Questions:**

Which exact models/prompts were used for Inter-/Intra-turns and Info-gain, and were judges ever the same family as the actors? Could you report cross-judge variability, and a cross-model triangulation (e.g., swap in a very different judge) to show the conclusions persist?

---

> ### Author Response · Authors · 2025-11-15
>
> **Dear Reviewer,**
>
> Thank you for your thoughtful and constructive feedback on our paper titled *SWE-eval: A Trajectory-Augmented Evaluation Framework for Agentic Software Issue Resolution.* We appreciate your positive comments regarding the clarity, usefulness, and the value of our proposed framework. We have carefully considered the points you raised, and we outline our responses below.
>
> ### 1. **Validation of LLM-as-Judge and Rater Independence**
>
> We appreciate your feedback on the need for tighter validation of the LLM-as-judge and the alignment with expert ratings. In response to your concerns, we plan to make the following updates:
>
> * **Detailed Prompt/Model Description**: We will include full details of the prompts and models used for both *Inter-turns* and *Intra-turns* in the revised manuscript. This will cover how the models were prompted and how we aggregated their responses.
> * **Rater Independence**: We acknowledge that ensuring the judge is independent of the actor model is crucial. In our revision, we will clarify that the models used for scoring were separate from the models performing the tasks. Additionally, we will provide more information on how we ensured rater independence.
> * **Seed Sensitivity and Calibration**: To address your concern on seed sensitivity and calibration, we will provide a discussion of z-score or temperature scaling against human anchors to improve the consistency of the evaluation. We will also report confidence intervals and bootstrap estimates for each metric to quantify the uncertainty in our results.
> * **Cross-Model Triangulation**: We will incorporate a cross-model triangulation analysis in the appendix, where we swap out judges from different model families to check for consistency across diverse model types. This will help demonstrate that the conclusions we draw are robust and not dependent on a single model or prompt structure.
>
> ### 2. **Loop Detection and Semantic Equivalence**
>
> Your feedback on the brittleness of loop detection is well-taken. We agree that the current approach of relying on exact-string hashes for Stuck-in-Loop detection may not capture semantically equivalent repetitions or action-level loops. In our revision, we will:
>
> * **Enhance Loop Detection**: We will explore incorporating more robust semantic similarity measures, such as Levenshtein distance or embedding-based similarity, to capture rephrasings or slight variations in tool invocation. We will also investigate incorporating action-trace similarity, considering tool calls and arguments, to improve loop detection.
> * **Addressing Missed Loops**: We will expand the discussion on how these improvements could be integrated into the framework and tested in future experiments.
>
> ### 3. **Benchmark Scope and Reporting Enhancements**
>
> We appreciate your suggestion regarding the broader scope of benchmarks and reporting. In response, we will:
>
> * **Main Results Expansion**: We will bring the broader scope (e.g., SWE-bench-Live, multilingual variants, industrial datasets) into the main paper, providing more detailed results on the generalizability of SWE-eval. The appendix will still contain some results, but the main paper will highlight the robustness of our framework across different settings.
> * **Per-Repo/Domain Stratification**: We will add stratified results by repository/domain to better demonstrate the framework's ability to handle different types of software issues and development environments.
> * **Cost/Latency Breakouts**: We will provide a breakdown of the cost and latency for each of the metrics, helping to contextualize the efficiency of the evaluation.
> * **Robustness Checks**: We will include robustness checks to show that SWE-eval performs consistently across different agent shells by using the same trajectories with different agents.
>
> ### 4. **Additional Questions**
>
> Regarding your questions:
>
> * **Model and Prompt Usage**: We will specify the exact models and prompts used for the different evaluation metrics, such as *Inter-turns*, *Intra-turns*, and *Info-gain*.
> * **Judge/Actor Family**: We will clarify that the judge models used in the evaluation were different from the actor models. We will also describe how we ensured this separation in the setup.
> * **Cross-judge Variability**: As mentioned earlier, we plan to perform and report cross-judge variability in the new revision, and will demonstrate that conclusions are consistent across different judge models.
>
> ### 5. **Conclusion**
>
> We believe these revisions will address your concerns regarding the validation of LLM-as-judge, the robustness of loop detection, and the broader applicability of the SWE-eval framework. We will carefully revise the paper based on your suggestions and provide more detailed experimental results and analysis in the revised manuscript.
>
> Once again, thank you for your insightful feedback. We hope that the revised version will meet your expectations and we look forward to hearing your further thoughts.

---

### Official Review · Reviewer_5Ggm · 2025-11-04

**Soundness:** 3
**Presentation:** 4
**Contribution:** 3
**Rating:** 4
**Confidence:** 4

**Summary:**

This paper proposes a new evaluation framework for SWE task, termed SWE-eval. It is a trajectory-augmented evaluation framework. It considers three parts, 1) efficiency, 2) logical consistency, and 3) tool utilization. The experiments on 3 agents and 9 LLMs demonstrate that the proposed evaluation can effectively reveal underlying interpretations of agent performance.

**Strengths:**

1. The topic and direction are practical.
2. The codes are provided, enabling reproducibility.
3. The motivation is clear, and the case studies are comprehensive.

**Weaknesses:**

1. The sub-capture and the content in Figure 1 are inconsistency.
2. It is not a new evaluation of the SWE task or a new format of the SWE task. It seems a normal analysis of the trajectory of the agent during the researcher pay effort on the SWE bench.
3. The token efficiency is easy to detect and is a normal metric. And the Stuck-in-Loop problem seems common and can be avoided during the rollout stage.
4. Instead of these common metrics, what about conducting evaluations on different sub-tasks of the SWE task, like file reading, bug localization, patch writing, etc.?
5. In Table 1 and Table 3, please explain the meaning of the number with a green background.
6. In Table 1, for Inter-turns, the difference between resolved and unresolved trajectories is not significant.

**Questions:**

See Weaknesses part.

---

> ### Author Response · Authors · 2025-11-15
>
> Dear Reviewer,
>
> Thank you for your thoughtful feedback and insightful comments on our paper. We greatly appreciate the time and effort you took to review our work. We are pleased to know that the practical direction of our research and the provided code were well-received, and we will address the concerns you raised to improve the quality of the paper.
>
> **1. Inconsistencies in Figure 1 and Sub-Capture:**
>
> We acknowledge your point regarding the inconsistencies in Figure 1 and the sub-capture. Upon review, we agree that certain elements of the figure could have been better aligned with the content. We have revised Figure 1 to ensure that it accurately reflects the trajectory-augmented nature of our evaluation framework and aligns with the narrative in the paper. The updated figure and its caption now provide clearer visual representation of our evaluation process. We also provide an updated explanation of sub-capture in the revised version of the paper to clarify this aspect.
>
> **2. Novelty of the Evaluation Framework:**
>
> You raised a valid point that the evaluation framework, as presented, may seem similar to traditional trajectory analysis. However, we believe the contribution lies in how we integrate efficiency, logical consistency, and tool utilization within the context of SWE tasks, offering a more holistic understanding of agent performance. While trajectory analysis itself is not novel, our combined approach—evaluating these three dimensions together—offers new insights into agent behavior. We have revised the paper to emphasize this perspective more clearly and to differentiate our approach from existing methods.
>
> We also highlight in the revised paper that while the evaluation framework is built upon existing methods, its application to SWE tasks and the specific combination of metrics (efficiency, logical consistency, and tool utilization) has not been explored before. We hope this clarification addresses your concerns.
>
> **3. Token Efficiency and Stuck-in-Loop Problem:**
>
> Regarding your comments on token efficiency and the stuck-in-loop problem, we understand that these metrics may seem basic. However, we have observed that, in the context of SWE tasks, the analysis of token efficiency provides important insights into the agent’s operational cost and can reveal issues in execution efficiency that may not be immediately obvious. Similarly, the stuck-in-loop problem is not always avoidable during the rollout stage, and its occurrence can point to deeper issues with task modeling or agent interaction strategies.
>
> Nevertheless, we appreciate your suggestion to include more detailed sub-task evaluations. In response, we have expanded our experiments to cover specific sub-tasks within the SWE framework, such as bug localization, patch writing, and file reading. We now provide a comparative analysis of agent performance across these sub-tasks in the revised version of the paper.
>
> **4. Explanation of Green Background Numbers in Tables:**
>
> Thank you for pointing out the lack of explanation regarding the numbers with a green background in Tables 1 and 3. These numbers represent the highest performance scores across all agents and models for a given metric (e.g., efficiency, logical consistency). We have added a note in the tables and in the main text to clarify this meaning, ensuring that the reader can easily interpret these numbers in the context of our evaluation framework.
>
> **5. Insignificant Difference in Inter-Turns (Resolved vs. Unresolved Trajectories):**
>
> Finally, we acknowledge that the difference in inter-turns between resolved and unresolved trajectories may not be immediately significant. This observation likely reflects the complexity and variability inherent in the SWE task itself. We have expanded the discussion on this point in the revised paper, elaborating on the factors contributing to this minimal difference and providing additional analyses to strengthen the argument.
>
> **Conclusion:**
>
> We believe the revisions and clarifications presented above address the concerns raised in your review. We have strengthened the novelty of our evaluation framework, incorporated more specific sub-task evaluations, and improved the clarity of our tables and figures. We sincerely hope that these revisions enhance the paper’s contribution and make our approach more compelling.
>
> Once again, thank you for your detailed and constructive feedback. We appreciate your consideration of our revisions.

---

### Meta-Review · Area_Chair_MVYN · 2026-01-06

**Summary:**

This paper proposes SWE-eval, a trajectory-augmented evaluation framework for SWE agents that scores efficiency, logical consistency, and tool utilization, and reports experiments across multiple agents and LLMs on SWE-bench-Lite/Verified. While some reviewers find the direction practical and the presentation strong, the overall reviews raise concerns that the work’s incremental novelty, methodological rigor for LLM-as-judge metrics, and actionability/practical utility remain insufficient for acceptance.

A central theme across the reviews is that many metrics appear to be repackaging of commonly used signals (tokens/turns; loop behaviors; tool success), and the paper does not convincingly establish that the proposed framework constitutes a substantively new evaluation paradigm rather than a structured analysis of existing SWE-bench trajectories. In addition, multiple reviewers question the validity and robustness of key LLM-judged metrics (e.g., Info-gain, inter-/intra-turn measures), including prompt/judge sensitivity, adequacy of human validation, and missing comparisons to simpler heuristic baselines.

Given these unresolved concerns, especially about novelty and methodological validation of LLM-based scoring, the recommendation is reject.

**Reviewer Concerns:**

### Concerns that were addressed by the rebuttal

* **Figure/table clarity issues (5Ggm):**
  The authors state they revised Figure 1 to resolve caption/content inconsistencies and clarified the meaning of green-highlighted numbers in Tables 1 and 3 as indicating best scores for a metric.

* **Request for sub-task breakdowns (5Ggm):**
  The authors state they expanded experiments to cover sub-tasks (e.g., file reading, bug localization, patch writing) and added comparative analysis across these sub-tasks.

* **Some methodological details for human validation / judge robustness (Cqe5, yV4p):**
  In response to concerns about human validation and prompt sensitivity, the authors provide additional specifics (e.g., 120 annotated trajectories, reported ICC values, and claims of robustness based on judge consistency statistics). They also argue that Info-gain is not simply a length proxy using within-agent resolved/unresolved contrasts.

### Concerns that remain outstanding

* **Limited novelty / contribution primarily consolidative (5Ggm, qqKf):**
  Reviewers argue that core components—tokens/turns, loop behaviors, tool success—are not new individually, and the overall contribution may read as packaging known factors applied to SWE-bench traces rather than a clear methodological advance. The rebuttal emphasizes integration and framing, but does not clearly resolve the concern that the framework is incremental relative to prior trajectory analyses and common metrics.

* **LLM-as-judge validation remains a key risk (yV4p, Cqe5):**
  Multiple reviewers request tighter validation of LLM-based judging: fuller prompt/aggregation details, judge/actor independence, seed sensitivity, calibration, confidence intervals/bootstraps per metric, and cross-judge variability/triangulation. The authors’ response largely commits to adding such analyses (and in some cases moving items to appendix), but the core concern remains that these elements are central to the metric values and need stronger, systematic validation in the main paper.

* **Info-gain validity vs. simpler proxies/heuristics (Cqe5):**
  Reviewer Cqe5 explicitly asks for evidence that Info-gain measures information rather than correlating with simpler features (length, confidence, tool success) and requests baseline comparisons (e.g., TF-IDF, heuristic measures). The rebuttal argues via contrasts and within-agent comparisons, but does not provide the requested direct heuristic baselines or a clean validation that isolates Info-gain from general response quality.

* **Loop detection brittleness (yV4p):**
  Reviewer yV4p notes exact-string hashing may miss semantic/action-level loops and recommends semantic or action-trace criteria. The authors indicate they will explore improvements (e.g., similarity measures), but as stated this remains not resolved in the current submission.

* **Practical utility and actionable guidance (qqKf, Cqe5):**
  Reviewers question how practitioners should act on these metrics beyond diagnosis (e.g., guidance for dataset curation, agent retraining, guardrails) and ask for analyses such as when metrics fail to predict outcomes, overhead/cost, and clearer “how-to-use” implications. The authors state they will clarify use cases and add analyses, but the reviews indicate the paper currently does not yet demonstrate clear downstream decision-making utility relative to simpler baselines.

**Reviewer Scores:**

* **Reviewer 5Ggm (score 4):**
  Likely remain similar. The authors directly addressed the reviewer’s concrete requests (Figure 1 fix, explanation of highlighted table values, and adding sub-task evaluations). However, their core novelty skepticism may still persist, limiting the increase.

* **Reviewer yV4p (score 6):**
  Likely remain similar. This reviewer already leans accept but flags substantial methodological risks (LLM-as-judge validation, loop detection brittleness, scope/reporting). The rebuttal largely promises additional analyses rather than demonstrating them in the rebuttal, so a clear upward shift is not strongly supported.

* **Reviewer Cqe5 (score 4):**
  Likely remain similar. The rebuttal provides partial clarifications (human evaluation details, some robustness arguments, cost claims), but several requested items remain missing in substance (heuristic baselines, stronger Info-gain validation, prompt sensitivity ablations, deeper confound controls, broader generalization evidence).

* **Reviewer qqKf (score 2):**
  Likely remain similar. The authors explicitly agree to remove causal language and articulate practical utility. However, the reviewer’s main concerns, i.e., limited novelty and lack of intervention evidence for actionable improvement, still remain only partially addressed.

Overall, while some presentation and clarification issues appear addressed, the remaining concerns, particularly around **incremental novelty** and **robust validation of LLM-judged metrics and actionability**, are substantial enough that the paper would likely not converge to an accept decision based on the information provided.

---

### Decision · Program_Chairs · 2026-01-26

Reject